FERMILAB-PUB-23-698-CSAID

# Towards a data-driven model of hadronization using normalizing flows

Christian Bierlich[1]♠, Phil Ilten[2]†, Tony Menzo[2,3,4]⋆, Stephen Mrenna[2,5]✠, Manuel Szewc[2]‖, Michael K. Wilkinson[2]⊥, Ahmed Youssef[2]‡, and Jure Zupan[2,3,4]§

[1] Department of Physics, Lund University, Box 118, SE-221 00 Lund, Sweden
[2] Department of Physics, University of Cincinnati, Cincinnati, Ohio 45221, USA
[3] Berkeley Center for Theoretical Physics, University of California, Berkeley, CA 94720, USA
[4] Theoretical Physics Group, Lawrence Berkeley National Laboratory, Berkeley, CA 94720, USA
[5] Scientific Computing Division, Fermilab, Batavia, Illinois 60510, USA

♠christian.bierlich@hep.lu.se, †philten@cern.ch, ⋆menzoad@mail.uc.edu, ✠mrenna@fnal.gov, ‖szewcml@ucmail.uc.edu, ⊥michael.wilkinson@uc.edu, ‡youssead@ucmail.uc.edu, §zupanje@ucmail.uc.edu,

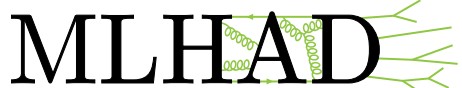

## Abstract

We introduce a model of hadronization based on invertible neural networks that faithfully reproduces a simplified version of the Lund string model for meson hadronization. Additionally, we introduce a new training method for normalizing flows, termed MAGIC, that improves the agreement between simulated and experimental distributions of high-level (macroscopic) observables by adjusting single-emission (microscopic) dynamics. Our results constitute an important step toward realizing a machine-learning based model of hadronization that utilizes experimental data during training. Finally, we demonstrate how a Bayesian extension to this normalizing-flow architecture can be used to provide analysis of statistical and modeling uncertainties on the generated observable distributions.

# 1  Introduction

Hadronization is one of the least understood ingredients in the simulation of particle collisions. While the Lund-string [1, 2] and cluster-fragmentation [3–5] models give reasonable overall descriptions of hadronization, there are still significant discrepancies between both the two models [6] and the models and data [7]. Augmenting these phenomenological models with a *data-driven* description of hadronization may help to improve the predictions.

Hadronization models, such as the string and cluster models, serve two distinct purposes. The first purpose is rooted in direct physics motivation. We aim to enhance our understanding of QCD behavior beyond the approximations afforded by lattice quantum chromodynamics (QCD) and perturbative QCD under specific limits. For this purpose, reliance on models is essential. The second purpose is to provide a realistic description of final state particles in high energy collisions. This description allows for the detailed study of detector responses, as well as realistic modeling of both background and signals for high momentum transfer processes. This modeling is critical in most high energy particle physics analyses both for interpreting the results as well as estimating systematic uncertainties.

In the first scenario, discrepancies between models offer an opportunity to utilize measurements to deepen our understanding of physics. In this context, substituting a physics model with a machine learned (ML) model may obscure these discrepancies or mask the foundational physics phenomena. However, a carefully designed ML model could also provide insights into the physics model by supplying a detailed description across all phase space of the physics model at a granular level. In the second scenario, discrepancies with data do not provide deeper insights into the hadronization process but rather produce more ambiguous interpretations of experimental data with larger associated systematic uncertainties. Here, data-driven models can reduce these uncertainties while retaining the same physics motivation present in the original models.

To develop these data-driven models the existing phenomenology can be augmented, keeping the underlying strings or clusters as the building blocks, but perturbing their dynamics to accommodate all relevant experimental observables. This is a problem well suited for ML methods, which can form a flexible basis for adjusting the underlying model dynamics to match experimental data. The first attempts at providing an ML description of simplified hadronizing systems have been carried out using both (MLHAD) conditional sliced Wasserstein autoencoders (cSWAEs) [8] and (HADML) generative adversarial networks (GANs) [9, 10]. These two architectures have reproduced key features of the Lund string model in PYTHIA 8 [11] and the cluster model in HERWIG 7 [12, 13], respectively, but both rely on training data that is not available at the experimental level. Here, ML-

HAD [8] uses the kinematics of first hadron emissions from a string, which is only available at the generator level. The HADML model uses either the same information from cluster decays [9] or the full hadron-level kinematic information for collisions [10], which is not yet available in practice. Similarly to the present paper, Ref. [10] does attempt to improve the agreement between predictions of ML based hadronization model with macroscopic observables, in a simplified set-up.

At present, there are three main challenges to performing ML training on real experimental data: (1) develop a procedure to alter microscopic string dynamics for parton systems produced from existing event generators; (2) quantify the uncertainties associated with this procedure and propagate them through detector and material simulations; and (3) identify and measure an adequately large set of observables sufficiently sensitive to hadronization to break model degeneracy. Here, we address the first two challenges.

Building on the work of the MLHAD model from ref. [8], we analyze a simplified version of the Lund string model, now in the context of normalizing-flow (NF) ML architectures [14–16]. The NF architectures transform a simple underlying probability density into the complex hadronization probability density via mappings that can be inverted. This specific feature of NFs is key to the work presented here; hadronization from one NF model can be reweighted to another model with minimal computational cost. The NF architectures presented here surpass the previous cSWAE architecture in both efficiency and physics capabilities, and enables, in the context of Bayesian NFs (BNFs) [17], a coherent analysis of uncertainties. Furthermore, it provides a method for determining microscopic dynamics from macroscopic observables via a novel training approach termed MAGIC.

This paper is organized as follows: in section 2 we briefly review the simplified Lund string model. This model is used in section 3 to train a NF architecture for hadronization. In section 4 we use a modified NF to introduce the MAGIC method for fine-tuning hadronization models, while in section 5 we use Bayesian NFs to estimate uncertainties. Finally, section 6 contains our conclusions. Appendices contain details about the public code, appendix A, and a pedagogical introduction to Bayesian normalizing flows, appendix B.

## 2 Hadronization

Hadronization describes the conversion of a partonic system consisting of quarks $q$, antiquarks $\bar{q}$, and gluons $g$ into a final state consisting of hadrons $h$. In what follows, we use the Lund string model of hadronization [1,2] to describe the simplest hadronizing partonic system, a $q_i\bar{q}_i$ pair of massless quarks with flavor $i$. Specifically, we consider the quark system in the center of mass frame, with the quark and antiquark in close proximity and traveling apart with equal and opposite momenta. In the Lund model, as the separation between the quark and antiquark increases, the non-Abelian nature of the strong force causes an approximately uniform string, or flux tube, of color field to form between the quark and the antiquark, with an approximately uniform energy density $\kappa \approx 1\,\text{GeV/fm} \approx 0.2\,\text{GeV}^2$. The quark and antiquark are the endpoints of this string.

The constant force between the quark and antiquark translates into a potential energy that increases with their separation. As the kinetic energy of the quark and antiquark at the endpoints of the string is converted into the potential energy of the string, it can become energetically favorable to create $q\bar{q}$ pairs out of the vacuum, thereby breaking the string. The original string breaks into fragments, e.g., a composite hadron $h \equiv q_i\bar{q}_j$ and a string fragment with endpoints $q_j\bar{q}_i$. Multiple emissions can be implemented sequentially by boosting and rotating into each hadronizing string fragment's center-of-mass frame, emitting a hadron while conserving energy and momentum, then boosting and rotating

the hadron and the new string fragment back to the rest frame of the initial string.

The kinematics of the emitted hadron is determined through a correlated sampling of transverse momentum $p_\perp$ and longitudinal momentum fraction

$$z \equiv (E \pm p_z)_{\text{hadron}}/(E \pm p_z)_{\text{string}} \, ,$$

where $E$ and $p_z$ are the energy and longitudinal momentum of the hadron or string, as labeled, in the center-of-mass frame of the string fragment from which the hadron is emitted, and the parton is moving in the $\pm \hat{z}$ direction in the same frame. After each hadron emission, the kinematics of the string are updated. Although the $q_j \bar{q}_j$ pair has no net transverse momentum, both the $q_j$ quark and $\bar{q}_j$ antiquark carry transverse momentum $p_\perp \equiv \sqrt{p_x^2 + p_y^2}$, where $p_x$ and $p_y$ are perpendicular to each other and sampled from a Gaussian probability distribution

$$\mathcal{P}(p_x, p_y; \sigma_{p_\perp}) = \frac{1}{2\pi\sigma_{p_\perp}^2} \exp\left(-\frac{p_x^2 + p_y^2}{2\sigma_{p_\perp}^2}\right) \, , \tag{1}$$

where the width parameter $\sigma_{p_\perp}$ is obtained from fits to data.[1]

The probability for a hadron to be emitted with longitudinal lightcone momentum fraction $z$ is given by the Lund symmetric fragmentation function

$$f(z) \propto \frac{(1-z)^a}{z} \exp\left(-\frac{bm_\perp^2}{z}\right) \, , \tag{2}$$

where $m_\perp^2 \equiv m^2 + p_\perp^2$ is the square of the transverse mass, $m$ is the hadron mass, and $a$ and $b$ are fixed parameters determined by fits to data.[2] Each iteration of causally disconnected string fragmentations consists of: randomly selecting one string end; assigning probabilistically a quark flavor to be pair produced during string breaking; generating the transverse momentum of this pair; generating the light-cone momentum fraction of the new hadron; and finally computing the longitudinal momentum of the new hadron, by conserving the total energy and momentum of the system. Iterative fragmentation continues until the energy of the string system crosses a threshold. The remaining string piece is then combined into a final pair of hadrons such that the energy of the initial two-parton system is converted entirely into the emitted hadrons.

Working within the Lund string model, the phenomenology of hadronization is largely determined by the probabilities with which different hadron species are produced, *i.e.*, (i) the forms of the probability distributions for the hadron momenta traditionally determined by eqs. (1) and (2), (ii) the method of determining the color singlet systems, and (iii) the process of flavor selection. Here, we set aside (ii) and (iii) to focus on (i), the kinematics of string fragmentations, for which we want to ultimately develop a *data-driven* determination of the probability distributions.

## 3   Normalizing flows

Normalizing flows (NFs) are generative ML models that can produce high-quality continuous approximations of probability distributions from a limited set of data samples [14–16].

---

[1]Within PYTHIA 8, $\sigma_{p_\perp}$ is set with the parameter name and default value of `StringPT:sigma = 0.335`.
[2]The default parameter names and values as implemented in PYTHIA 8 are `StringZ:aLund = 0.68` and `StringZ:bLund = 0.98`, for $a$ and $b$, respectively.

They accomplish this by concatenating a series of $N$ independent, bijective transformations, $F(\boldsymbol{z}) = f_N(f_{N-1}(\ldots f_2(f_1(\boldsymbol{z}))\ldots))$, which then map a latent space probability distribution $\mathcal{P}_Z(\boldsymbol{z})$ to a target distribution $\mathcal{P}_X(\boldsymbol{x})$. The latent space is typically chosen such that it can be easily sampled. The form of the bijective functions $f_i$ is adjusted by modifying the model parameters $\boldsymbol{\theta}$ and any external parameters, including any provided conditional labels $\boldsymbol{c}$.

Since each $f_i$ is a continuous function, the composite function $F$ is also continuous. This allows for density estimation over the full phase space including regions sparsely populated by the training data. In section 4 we use this feature to introduce a method for fine-tuning the form of the microscopic fragmentation function using measured observable quantities. Furthermore, in the MLHAD NF architecture we use Bayesian NFs (BNFs) [17], in which the model parameters $\boldsymbol{\theta}$ themselves are random variables. They are taken to be normally distributed, with average values and variances learned from training data and which encode data uncertainties, as described in section 5. Further details on both NFs and BNFs are given in appendix B.

The MLHAD NF architecture is able to reproduce pseudo-data generated using a simplified version of the PYTHIA 8 Lund string hadronization model. This pseudo-data is produced using the same simplified model as in ref. [8], in which only light-quark flavors are allowed as endpoints, and isospin symmetry is required, *i.e.*, only neutral and charged pions at a single mass are generated. The MLHAD NF is trained on a dataset of $N$ hadron emissions from a string with energy $E_{\mathrm{ref}} = 200\,\mathrm{GeV}$. That is, the training dataset consists of $N$ two-dimensional arrays of first hadron emission kinematics $\boldsymbol{x}_n = \{p_{z,n}, p_{\perp,n}\}$, where $n \in \{1, \ldots, N\}$ and $p_z$ and $p_\perp$ are, respectively, the longitudinal and transverse components of the emitted hadron's momentum in the-center-of-mass frame of the string. To generate hadron kinematics for strings with energies other than $E_{\mathrm{ref}}$, we use the rescaling property of the Lund string fragmentation function to render $p_z$ independent of the string energy, transforming the generated value of $p_z$ according to $p_z \to p_z E_{\mathrm{ref}}/E$, where $E$ is the energy of the quark in the initial string's center-of-mass frame [8].

Unlike in ref. [8], we train the MLHAD NF on a dataset containing different transverse masses, $m_\perp$. For this, we construct labeled training datasets $\{x_n, c_n\}_{n=1}^N$, where

$$c_n \equiv \frac{m_{\perp,\mathrm{max}} - m_{\perp,n}}{m_{\perp,\mathrm{max}} - m_{\perp,\mathrm{min}}}\,, \tag{3}$$

$m_{\perp,\mathrm{min}} = m_{\pi^\pm} \approx 0.140\,\mathrm{GeV}$ and $m_{\perp,\mathrm{max}} = 1.3\,\mathrm{GeV}$ are, respectively, the minimal and maximal values of $m_\perp$ used in training. The maximum is chosen such that none of the hadronization chains considered will produce an $m_\perp$ above this value. That is, the conditional labels $c_n$ are functions of hadronic transverse mass $m_\perp$ such that $c_n \in [0, 1]$, where the boundaries correspond to the minimum and maximum $m_\perp$. Here, $m_\perp$ is used rather than mass to ensure the independence of the $z$ and $p_\perp$ probability distributions for a given $m_\perp$ value, see eq. (2). The training dataset is split into 15 different conditional labels, where each label corresponds to a different fixed $m_\perp$. For each conditional label $5 \times 10^5$ first hadron emissions are used, for a total of $N = 7.5 \times 10^6$ emissions in the full training dataset.

Figure 1 shows a comparison between PYTHIA 8 generated kinematic distributions and the learned MLHAD NF kinematic distributions for different values of the transverse mass (we have set the NF model parameters $\boldsymbol{\theta}$ to their fixed average values). We observe that the NF model can fully reproduce the $p_z$ of the hadronizing $q\bar{q}$ system of the Lund string model for arbitrary hadron mass. The result of a full hadronization chain, where hadrons are sequentially emitted from the string fragments, is shown in fig. 2; the string fragmentation terminates at $E_{\mathrm{cutoff}} = 25\,\mathrm{GeV}$ in this case in order to avoid the final combination step,

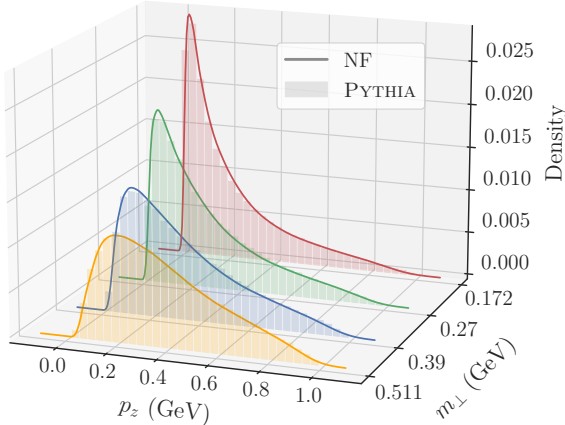

Figure 1: A comparison between the (histograms) Pythia 8 and (solid lines) MLhad NF generated single emission $p_z$ distributions produced at four different fixed values of $m_\perp$ which were not used in the training of the model.

which requires a separate treatment. We observe excellent agreement between the hadron multiplicities produced by Pythia 8 and the MLhad NF.

Next, we show how the MLhad NF model trained on single hadron emission pseudo-data can be used to adjust the microscopic model of single hadron emission kinematics, so that it reproduces experimental data that has no direct single hadron emission measurements.

# 4   Fine-tuning microscopic fragmentation kinematics

In this section we introduce a training method for fine tuning NF-based models of hadronization. The method, termed *m*icroscopic *a*lterations *g*enerated from *i*nfrared *c*ollections[3] (MAGIC), allows for the fine tuning of *microscopic* dynamics to describe a set of *macroscopic* observables. Practically, the microscopic dynamics are produced from an underlying phenomenological model, while the macroscopic observables are from experiment.

MAGIC is a natural extension of the traditional tuning techniques, such as manual tuning [18] or automated regression techniques [19–22]. Crucially, while approaches such as deep neutral networks using classification for tuning and reweighting (DCTR) [22] do parameter reweighting and tuning, they do not directly modify the underlying parametric Lund model used for training. MAGIC is able to increase the flexibility of the model beyond the parametric form, eqs. (1) and (2), while remaining physically meaningful by keeping the emission-by-emission paradigm described in section 2.

This method works by adding data-driven perturbations to an analytic solution, the Lund symmetric fragmentation function of eq. (2) in the case of hadronization, by augmenting it with an over-parameterized function such as an NF that can be modified arbitrarily to accommodate data. The Lund symmetric fragmentation function already provides a good description of experimental data; we seek to learn data-driven perturbations to obtain even better agreement with experiment.

As a toy example we take a simplified one-dimensional NF model, consisting of a weighted mixture of Gaussian distributions, trained on the $z$ component of the momentum

---

[3]Here, infrared collections refer to any hadronization-sensitive high-level observable distribution that can be obtained from experiment.

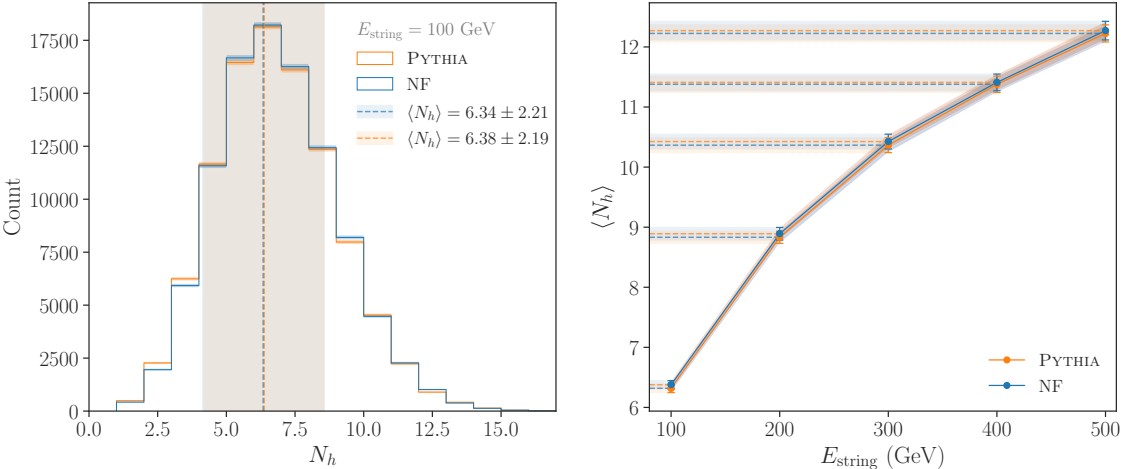

Figure 2: (left) Comparison of hadron multiplicity distributions generated with (orange) PYTHIA 8 and the (blue) MLHAD NF, constructed from hadronizations of $10^5$ strings, all with an initial energy of $E_{\text{string}} = 100\,\text{GeV}$ and an energy cutoff of $E_{\text{cutoff}} = 25\,\text{GeV}$. (right) Scaling of average hadron multiplicity $\langle N_h \rangle$ is given as a function of the starting string energy. Each marker represents an averaging of hadron multiplicity over the hadronization of $5 \times 10^3$ strings. The dotted lines show the average multiplicity $\langle N_h \rangle$ and the bands the corresponding $1\sigma$ range.

of first-emission hadrons, as described in section 3. MAGIC consists of two training phases: in the first phase, the NF is trained on simulated kinematics as described in section 3; in the second phase, the NF is modified to match the experimental data. The initial NF model, or base model, provides high fidelity sampling of single-hadron emission kinematics $\boldsymbol{x} = p_z$. Here we omit $p_\perp$, in contrast to section 3, for simplicity of the model. From these $\boldsymbol{x}$ kinematics, one can obtain predictions for measurable, hadronization-sensitive observables $\boldsymbol{y}$, *e.g.*, hadron multiplicity. In the second phase of training, the base model is fine-tuned by reweighting the dataset of $\boldsymbol{y}$ values generated by the base model to statistically match the experimentally observable dataset. This second phase explicitly relies on the ability to reweight distributions.

The training data for the second phase of MAGIC consists of three components: (i) the hadronization-chain-level kinematics $\boldsymbol{x}$, *i.e.*, the hadron kinematics $p_z$ from simulated emissions produced by the base model; (ii) the desired measurable observables from simulated hadronization chains produced by the base model $\boldsymbol{y}_{\text{sim}}$, *e.g.*, the hadron multiplicity $N_h$ predicted by the base model; and (iii) values of the same observables measured experimentally $\boldsymbol{y}_{\text{exp}}$. As a proof of principle, we use just a single observable, the total number of hadrons for a single hadronization chain, *i.e.*, the hadron multiplicity such that $\boldsymbol{y}$ is $N_h$.

An example of the training data, consisting of $N$ hadronization chains, is therefore[4]

$$
\boldsymbol{x} = \begin{pmatrix} \boldsymbol{x}_1 = \{p_{z,h_1}, p_{z,h_2}, p_{z,h_3}\}_1 \\ \boldsymbol{x}_2 = \{p_{z,h_1}, p_{z,h_2}, p_{z,h_3}, p_{z,h_4}\}_2 \\ \vdots \\ \boldsymbol{x}_N = \{p_{z,h_1}, p_{z,h_2}\}_N \end{pmatrix}, \quad \boldsymbol{y}_{\text{sim}} = \begin{pmatrix} \boldsymbol{y}_1 = N_{h,1} = 3 \\ \boldsymbol{y}_2 = N_{h,2} = 4 \\ \vdots \\ \boldsymbol{y}_N = N_{h,N} = 2 \end{pmatrix}. \tag{4}
$$

The fine tuning modifies the hadronization model, *i.e.*, the distribution governing $p_z$ for each emission, to minimize the difference between the two ensembles $\boldsymbol{y}_{\text{sim}}$ and $\boldsymbol{y}_{\text{exp}}$. We

---

[4]While not denoted explicitly in eq. (4), each array $\boldsymbol{x}_i$ is zero-padded to a fixed length of size $\max(\boldsymbol{y}_{\text{sim}}) = \max(N_{h,n})$.

do not match specific measured hadronization chains to a given hadronization history but instead compare the two ensembles at the statistical level.

MAGIC does not require regenerating hadron emissions for each perturbation of the NF, and instead reweights the hadronization chains, which is computationally advantageous. We make use of the fact that NFs give explicit access to the model likelihood and reweight events that were originally sampled from the base model to events sampled from the updated model. Each hadronization-chain weight can be computed in terms of the likelihood ratio between the updated, or perturbed, model likelihood $\mathcal{P}_X(\boldsymbol{x}_n, \boldsymbol{\theta}_P)$, and the base model likelihood $\mathcal{P}_X(\boldsymbol{x}_n, \boldsymbol{\theta}_B)$. Written in terms of single emissions, the likelihood $\mathcal{P}_X(\boldsymbol{x}_n, \boldsymbol{\theta})$ can be factorized as,

$$\mathcal{P}_X(\boldsymbol{x}_n, \boldsymbol{\theta}) = \prod_{i=1}^{N_{h,n}} \mathcal{P}_X(\boldsymbol{x}_{n,i}, \boldsymbol{\theta}) \,, \tag{5}$$

where $N_{h,n}$ is the number of hadrons in hadronization chain $n$, and $\boldsymbol{x}_{n,i}$ is emission $i$ of chain $n$.

We introduce a hadronization-chain weight array $\boldsymbol{w}$, where each weight is computed as the product of the likelihood ratios for all emissions in a chain

$$\boldsymbol{w} = \begin{pmatrix} w_1 \\ w_2 \\ \vdots \\ w_N \end{pmatrix}, \text{ where } w_n = \prod_{i=1}^{N_{h,n}} \frac{\mathcal{P}_X(\boldsymbol{x}_{n,i}, \boldsymbol{\theta}_P)}{\mathcal{P}_X(\boldsymbol{x}_{n,i}, \boldsymbol{\theta}_B)} \,. \tag{6}$$

Explicitly for this example, $\boldsymbol{x}_{n,i}$ is just the $p_z$ of hadron $i$ from hadronization chain $n$. The reduction in training time associated with this use of hadronization-chain weights is crucial for the technical feasibility of the MAGIC approach to fine-tuning.

The learning objective of the fine-tuning phase is to minimize the statistical distance between $\boldsymbol{y}_{\text{sim}}$, weighted by $\boldsymbol{w}$, and the target distribution $\boldsymbol{y}_{\text{exp}}$. In our toy example, we use the Wasserstein distance [23–26], or Earth mover's distance (EMD), as a measure of the similarity between the two samples[5] and define the loss function as

$$\mathcal{L}_{\text{EMD}}(\boldsymbol{y}_{\text{sim}}, \boldsymbol{y}_{\text{exp}}) = \sum_{n=1}^{N} \sum_{m=1}^{M} f_{n,m}^* d_{n,m} \,, \tag{7}$$

where the elements of the flow matrix $f_{n,m}$ encode the fractional amount of weight to be transferred between event $\boldsymbol{y}_{\text{sim},n}$ and $\boldsymbol{y}_{\text{exp},m}$ and $d_{n,m} = ||\boldsymbol{y}_{\text{sim},n} - \boldsymbol{y}_{\text{exp},m}||_2$ is the distance between these two hadronization chains. Here, $M$ is the number of hadronization chains observed in the experimental dataset.

Once the loss has been computed, back-propagation algorithms update the NF parameters $\boldsymbol{\theta}$ using PyTorch's automatic differentiation engine `autograd`. The `autograd` engine utilizes differential programming paradigms with dynamic computational graphs to yield the gradients of the loss function with respect to all parameters $\nabla_{\boldsymbol{\theta}} \mathcal{L}_{\text{EMD}}$ by tracking the impact of the hadronization-chain weights $\boldsymbol{w}(\boldsymbol{\theta})$ on the loss. We can then find a model likelihood that produces the targeted observable distribution because updating $\boldsymbol{\theta}$ corresponds to updating every $\mathcal{P}_X(\boldsymbol{x}_{n,i}, \boldsymbol{\theta})$. The only dynamical object in the fine-tuning phase of MAGIC is the hadronization-chain weight array $\boldsymbol{w}$; the base model, $\boldsymbol{x}$, $\boldsymbol{y}_{\text{sim}}$, and $\boldsymbol{y}_{\text{exp}}$ all remain fixed.

---

[5]In many cases, when training from real experimental distributions, one may only have access to binned datasets. The MAGIC paradigm may equivalently be used in these scenarios by simply utilizing a binned statistical distance such as $\chi^2$.

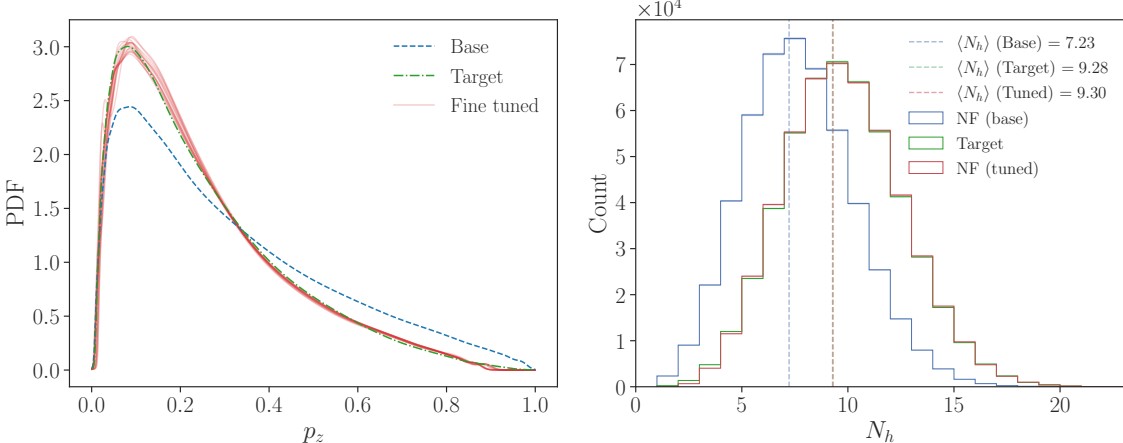

Figure 3: (left) Comparison between the single emission kinematic distribution $p_z$ of the (blue dashed) base model, the (green dashed-dotted) distribution used to generate $\boldsymbol{y}_{\mathrm{exp}}$, and (red) an ensemble of solutions learned using MAGIC. (right) Comparison of hadronization-chain level hadron multiplicity between the (blue) base, (red) one of the fine tuned, and the (green) target distributions where each histogram is constructed from $5 \times 10^5$ hadronization chains. The mean of each histogram is shown as a vertical dashed line.

In our toy example, we use a one-dimensional NF base model, see appendix B.1, trained on $N = 5 \times 10^5$ PYTHIA 8 generated hadronization chains with the Lund string parameter $a$ set to 0.68. Each of the transverse momentum components of the emitted hadrons is sampled from a Gaussian distribution, and the correlation between $p_z$ and $p_\perp$ is neglected for simplicity. We create a pseudo-experimental hadron multiplicity observable $\boldsymbol{y}_{\mathrm{exp}}$ with $M = 5 \times 10^5$ samplings of a second NF trained on PYTHIA 8 hadronization chains, produced with $a = 1.5$.[6]

We then perform 15 independent MAGIC fine-tunings[7] of the base model, where each fine-tuning is trained over three epochs in batches of $10^4$ samples with the learning rate fixed to $\delta = 2.5 \times 10^{-2}$ in the first epoch, and reduced to $\delta = 1 \times 10^{-2}$ for the remaining two epochs. The results of the independent trainings can be seen in the left panel in fig. 3; we see that the fine-tuned models all successfully learn the softer hadron emission spectrum used to generate the pseudo-experimental values of hadron multiplicity. In the right of fig. 3, we see that the corresponding hadron multiplicity distributions also agree.

We note some observations regarding the application of MAGIC. First, the fine-tuned models of $p_z$ shown in fig. 3 do not exactly match the target, and are not unique. This is expected, and is ultimately a consequence of finite training data and time. In multidimensional MAGIC fine-tunings with just a single observable, we found that qualitatively different solutions are obtained, all of which are able to reproduce the observable distributions matching those of the target. This degeneracy is presumably broken once additional sufficiently-orthogonal observables are included in $\boldsymbol{y}$.

The MAGIC fine-tuned models could be included within existing event generation pipelines, either directly as a kinematic generator or as a reweighter that utilizes the learned likelihood ratio between the base, *e.g.*, the default PYTHIA 8, and fine-tuned models, to provide

---

[6]We do not use PYTHIA 8 directly to generate the targeted pseudo-experimental multiplicity training dataset due to the included correlations between $p_z$ and $p_\perp$, unlike the simplified base model.

[7]Training using MAGIC is computationally inexpensive, for example, the training for the presented toy example can be performed on a modern laptop CPU with training times of $\mathcal{O}(1\ \mathrm{hour})$ to achieve similar accuracy to the results shown in fig. 3.

event weights similar to those of ref. [27]. We leave a full exploration and incorporation of the MAGIC method within PYTHIA 8 for future work.

# 5   Uncertainty estimation with Bayesian normalizing flows

When generating binned distributions using an ML model, there are typically two sources of uncertainty to consider: an uncertainty $\sigma_{\mathrm{gen}}$ due to the limited statistics of the generated data, as well as the systematic uncertainties due to the ML model. The systematic uncertainties can be further separated into $\sigma_{\mathrm{train}}$ and $\sigma_{\mathrm{data}}$, where $\sigma_{\mathrm{train}}$ captures the uncertainties due to the size of the training dataset and the ML architecture, while $\sigma_{\mathrm{data}}$ are the additional systematic uncertainties inherent to the training data. For example, when training on experimental data, the statistical uncertainty of that data contributes to $\sigma_{\mathrm{train}}$, while the systematic uncertainties, such as the detector resolution, contribute to $\sigma_{\mathrm{data}}$. While $\sigma_{\mathrm{gen}}$ is just the statistical uncertainty of the generated sample, $\sigma_{\mathrm{train}}$ and $\sigma_{\mathrm{data}}$ are typically much more difficult to quantify. Here, we propose methods to evaluate both $\sigma_{\mathrm{train}}$ and $\sigma_{\mathrm{data}}$.

Using a Bayesian NF, $\sigma_{\mathrm{gen}}$ and $\sigma_{\mathrm{train}}$ can be evaluated simultaneously, see appendix B.4. In this framework, the posterior of the network parameters $\boldsymbol{\theta}$ should capture how the training uncertainties result in different neural network choices that are all compatible with the training data. We first demonstrate that BNFs can capture both $\sigma_{\mathrm{gen}}$ and $\sigma_{\mathrm{train}}$ by training a BNF on a dataset of $N_{\mathrm{train}} = 10^5$ two-dimensional vectors of first-emission hadron kinematics given by $\{p_z, p_\perp\}$. Then, $M = 5 \times 10^4$ sets of randomly-selected network parameters $\boldsymbol{\theta}_m$ are sampled from the BNF posterior. For each $\boldsymbol{\theta}_m$, we generate an independent dataset of $N_{\mathrm{gen}} = 10^5$ first emissions.

We compute the average number of emissions $\langle N_{\mathrm{bin}} \rangle$ in each bin of hadron $p_z$, and its variance $\sigma_{\mathrm{bin}}$ across all $M$ BNFs:

$$\langle N_{\mathrm{bin}} \rangle \equiv \frac{1}{M} \sum_{m=1}^{M} \langle N_{\mathrm{bin}} \rangle_{\boldsymbol{\theta}_m} ,$$

$$\sigma_{\mathrm{bin}}^2 \equiv \langle N_{\mathrm{bin}} \rangle + \frac{1}{M} \sum_{m=1}^{M} \left( \langle N_{\mathrm{bin}} \rangle_{\boldsymbol{\theta}_m} - \langle N_{\mathrm{bin}} \rangle \right)^2 , \qquad (8)$$

where $\langle N_{\mathrm{bin}} \rangle_{\boldsymbol{\theta}_m}$ is the expected number of first emissions that fall in this particular $p_z$ bin, estimated from the sample of $N_{\mathrm{gen}}$ emissions generated using the NF with parameters $\boldsymbol{\theta}_m$. The left panel in fig. 4 compares the $\langle N_{\mathrm{bin}} \rangle$ from the learned model with the training dataset. We observe that for most bins, the model and the training dataset are consistent within uncertainty, although this degrades for sparsely populated bins where the model has not been trained with sufficient data.

The right panel in fig. 4 compares $\sigma_{\mathrm{gen}} = \sqrt{\langle N_{\mathrm{bin}} \rangle}$ with the total uncertainty $\sigma_{\mathrm{bin}}$ in a given bin. The total uncertainty includes both the generated uncertainty as well as the training uncertainty, $\sigma_{\mathrm{bin}}^2 = \sigma_{\mathrm{gen}}^2 + \sigma_{\mathrm{train}}^2$. For illustrative purposes, we compare in fig. 4 (right) the relative uncertainties in each bin, $\sigma_{\mathrm{bin}}/\langle N_{\mathrm{bin}} \rangle$ and $\sigma_{\mathrm{gen}}/\langle N_{\mathrm{bin}} \rangle$, rather than the absolute ones. The BNF estimate of the total uncertainty $\sigma_{\mathrm{bin}}$ is always larger than $\sigma_{\mathrm{gen}}$ because it contains also $\sigma_{\mathrm{train}}$, the additional model uncertainty due to finite training statistics and model choice. We observe that in the sparsely populated bins the relative uncertainty is large, with $\sigma_{\mathrm{bin}}$ significantly larger than $\sigma_{\mathrm{gen}}$, signaling that the BNF model is a poor predictor in this kinematic regime, primarily due to the lack of training data in the corresponding bins. Conversely, in densely populated bins the model is an accurate approximation of the data and the estimated BNF uncertainty $\sigma_{\mathrm{bin}}$ approaches

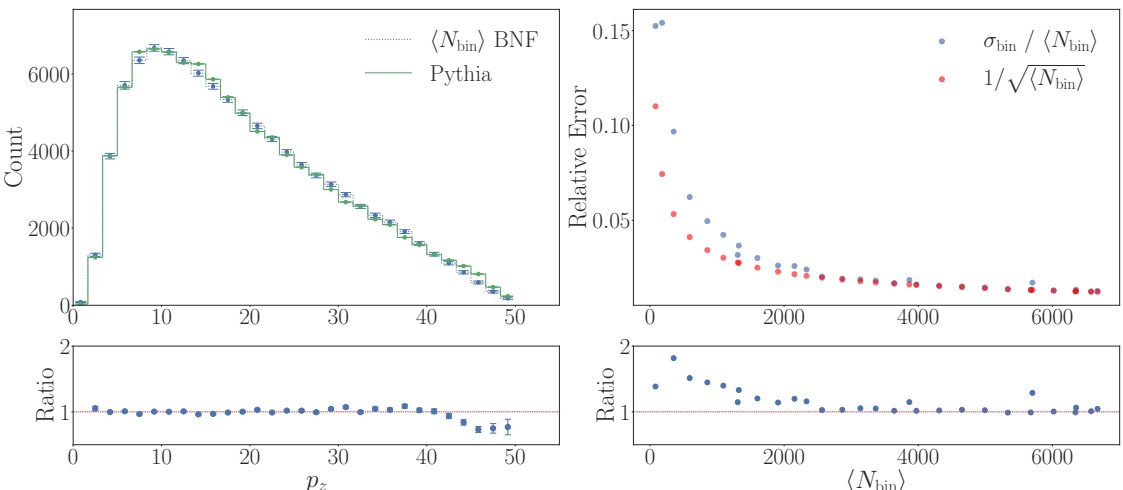

Figure 4: Generated and training uncertainties captured by the BNF. (left) Comparison of the $p_z$ distribution generated by (green) PYTHIA 8 and the (blue) MLHAD BNF. The PYTHIA 8 distribution was generated with $10^5$ single emissions, and the uncertainty is given by $\sqrt{N_{\text{bin}}}$. The BNF distribution is generated from an ensemble average over $5 \times 10^4$ BNFs, each consisting of $10^5$ single emissions, with means and uncertainties calculated using eq. (8). (right) Comparison of the (blue) BNF estimate of the relative uncertainty, $\sigma_{\text{bin}}/\langle N_{\text{bin}} \rangle$, and the (red) relative generated uncertainty, $\sigma_{\text{gen}}/\langle N_{\text{bin}} \rangle$, for each bin.

$\sigma_{\text{gen}}$. This implies that either the posterior is relatively certain of the underlying probability distribution, or that the model has been over-fit. To avoid over-fitting, we regularized the training to stop when the model performance on a validation dataset does not improve after 50 epochs.

A more detailed study of the uncertainties as a function of the number of generated events $N_{\text{gen}}$ is presented in fig. 5 for two representative $p_z$ bins from fig. 4: the densely populated bin, $p_z \in [6.67, 8.33)$ GeV and the scarcely populated bin, $p_z \in [48.33, 50)$ GeV. The expected values $\sigma_i$ for each $N_{\text{gen}}$ in fig. 5 were obtained by first computing $\langle N_{\text{bin}} \rangle_{\boldsymbol{\theta}_m}$ by approximating the integral in eq. (36) of appendix B.4 with a very large number of emissions, $10^7$, and then using $\langle N_{\text{bin}} \rangle_{\boldsymbol{\theta}_m}$ to obtain $\sigma_i$ as in eq. (8). This strategy ensures that the integration uncertainty is negligible and independent of $N_{\text{gen}}$. For the largest bin $\sigma_{\text{gen}}$ dominates even when $N_{\text{gen}} > N_{\text{train}}$ until for sufficiently large $N_{\text{gen}}$, $\sigma_{\text{train}}$ becomes the leading uncertainty on the generated dataset. This is expected, since the model is acting as a fit and, if sufficiently constrained, need not return the statistical uncertainty of the bin counts in the training data. For densely populated bins, $\sigma_{\text{train}}$ may thus be smaller than just the statistical uncertainty in a particular bin of the training dataset. In contrast, for the smallest bin, $\sigma_{\text{train}}$ is the dominant uncertainty even when $N_{\text{gen}} \lesssim N_{\text{train}}$, indicating that the model is a poor predictor with the posterior yielding a larger variance for $\boldsymbol{\theta}$.

The NF architecture of section 3 can also be used to efficiently assess the systematic uncertainties of experimental data, $\sigma_{\text{data}}$, as long as the NF is conditioned on these systematic uncertainties during training. Here, we consider a toy example where we treat the uncertainties on the parameters of the hadronization model as systematic uncertainties. Specifically, we introduce an uncertainty on the parameter $b$ of the PYTHIA 8 Lund string model, see eq. (2). We train the BNF on pseudo-data produced by PYTHIA 8, where the value of $b$ is set to three discrete values encoded by the conditional labels: the base value $b_B = 0.98$ and two perturbed values $b_P \in \{0.8, 1.4\}$, corresponding to a systematic

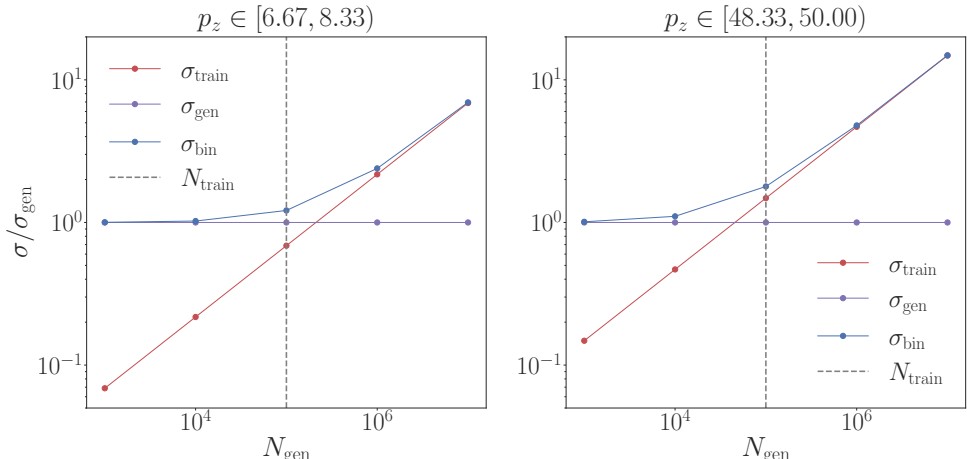

Figure 5: Study of the component uncertainties obtained with the BNF: the generated uncertainty $\sigma_{\text{gen}}$, training uncertainty $\sigma_{\text{train}}$, and the total uncertainty $\sigma_{\text{bin}}$ evaluated by the BNF for the (left) most populated and (right) least populated bins of the $p_z$ distribution in fig. 4.

uncertainty envelope on $b$.

After training, a large dataset of hadronization chains can be generated with this BNF using the conditional label $b_P$. However, since the probability for each hadron emission, $\boldsymbol{x}_n \sim \mathcal{P}_X(\boldsymbol{x}_n, b)$, is now a known function of $b$ from the conditioned training, we can also calculate and track the probability for each perturbed conditional label $b_P$ per hadron emission. The weight for a hadronization chain to be produced with $b_P$ rather than $b_B$ is

$$w \approx \prod_{n=1}^{N_h} \frac{\mathcal{P}_X(\boldsymbol{x}_n, \boldsymbol{\theta}^*, b_P)}{\mathcal{P}_X(\boldsymbol{x}_n, \boldsymbol{\theta}^*, b_B)} \, , \tag{9}$$

where $N_h$ is the number of hadrons emitted in the particular hadronization chain, *i.e.*, the hadron multiplicity for the hadronization of that particular string. We can then quickly produce large datasets for the values of $b_P$, by simply reweighting the initial dataset generated with $b_B$.

It is important to note that eq. (9) would be an equality, rather than an approximate relation, had we marginalized over all possible network parameters $\boldsymbol{\theta}$. However, for the sake of expediency, we use a fixed set $\boldsymbol{\theta}^*$ instead. Although $\boldsymbol{\theta}^*$ can be chosen to be the parameter set that provides a maximum a posteriori (MAP) value, in this example we simply sample an arbitrary $\boldsymbol{\theta}^*$ per emission for the posterior distribution of the BNF.

In fig. 6 we illustrate the reweighting process by plotting the multiplicity distribution, the number of hadrons produced per hadronization chain, from a large sample of hadronization chains produced by our BNF. This distribution is then reweighted to the two perturbed values of $b$. We see that reweighting each hadronization chain using (9) leads to multiplicity distributions that match the Pythia 8 generated ones within statistical uncertainty. Beyond this toy example, eq. (9) can be used to reweight any sample generated by a BNF to variations of the systematic uncertainties, assuming that the BNF is initially trained with conditional labels for each systematic uncertainty to be considered.

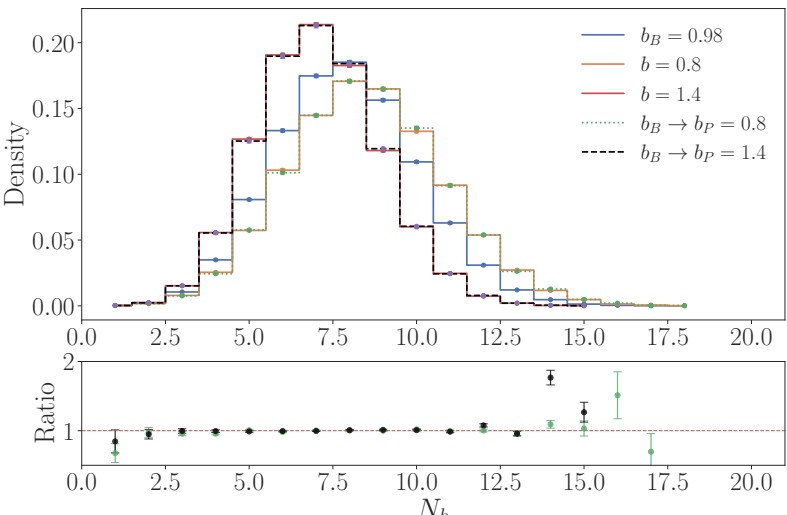

Figure 6: Distribution of the number of hadrons produced per hadronization chain for a sample of $5 \times 10^4$ strings. These hadronization chains are generated using three BNFs with fixed network weights conditioned on three distinct datasets: (blue) the base dataset $b_B = 0.98$, (orange) a perturbed dataset with $b_P = 0.8$, and (red) a perturbed dataset with $b_P = 1.4$. The distribution from the BNF for the base dataset is then reweighted using eq. (9) to the perturbed values of $b$, (green) $b_P = 0.8$ and (blue) $b_P = 1.4$.

## 6  Conclusions

In this manuscript, we have shown that the normalizing flow architecture is well suited for modeling the non-perturbative process of hadronization. A key feature of the normalizing flows is the analytic knowledge of the probability distribution for individual hadron emissions. The architecture is able to generate high fidelity first hadron emission kinematic samples, and Bayesian normalizing flows can be used to provide estimates of the hadronization uncertainty.

We have also introduced a novel training method, MAGIC, which allows for the systematic alteration of microscopic fragmentation dynamics such that the predictions best fit macroscopic observables, *e.g.*, hadron multiplicities. The MAGIC method avoids the need to produce additional samples each time the symmetric Lund string fragmentation is modified, and calculating instead appropriate weights for existing data samples. The use of automatic differentiation to update the model parameters makes the training numerically efficient.

We have showcased the potential of both Bayesian normalizing flows and MAGIC in the context of hadronization using toy examples, but there are a number of steps that still need to be made before data can be used in training, such as considering gluon-strings and flavor selection, both of which are part of ongoing work.

**Acknowledgments.** AY, JZ, MS, and TM acknowledge support in part by the DOE grant DE-SC1019775, and the NSF grant OAC-2103889. JZ acknowledges support in part by the Miller Institute for Basic Research in Science, University of California Berkeley. SM is supported by the Fermi Research Alliance, LLC under Contract No. DE-AC02-07CH11359 with the U.S. Department of Energy, Office of Science, Office of High En-

ergy Physics. CB acknowledges support from the Knut and Alice Wallenberg foundation, contract number 2017.0036. PI is supported by NSF grant OAC-2103889 and NSF-PHY-2209769. TM acknowledges support in part by the U.S. Department of Energy, Office of Science, Office of Workforce Development for Teachers and Scientists, Office of Science Graduate Student Research (SCGSR) program. The SCGSR program is administered by the Oak Ridge Institute for Science and Education for the DOE under contract number DE-SC0014664.

## A  Public code

The public code can be found at https://gitlab.com/uchep/mlhad in the BNF/ subdirectory. The repository consists of a hierarchical structure with three major components. The first component contains the implementation of the one- and two-dimensional NF network, with and without conditioning, the second component constitutes the integration of these NFs into fragmentation chains, and the third component is the implementation of MAGIC. Detailed explanations and examples of each component can be found within the code documentation and example notebooks. All code is written in Python, developed using v3.11, and heavily utilizes the PyTorch, developed using v2.1, deep learning library. Finally, all training datasets were produced using Pythia v8.309.

## B  Further details on normalizing flows

In this appendix we give further details on the Bayesian normalizing flows that we use in MLhad. Appendix B.1 reviews the basics of normalizing flows, appendix B.2 contains a brief review of Bayesian neural networks, and Section B.3 describes Bayesian normalizing flows.

### B.1  Normalizing flows

Normalizing flows (NFs) [14–16] are a class of generative ML architectures that can produce high fidelity continuous approximations of complex probability distributions using a finite collection of data samples. This is achieved by learning a composition of $n$ independent bijective transformations that relate a probability distribution $\mathcal{P}_Z(\boldsymbol{z})$ on a chosen latent space $Z$ to the target distribution $p_X(\boldsymbol{x})$ on target space $X$.

More precisely, given a multivariate random variable $\boldsymbol{z} \in \mathbb{R}^d$ and an invertible map $f : \mathbb{R}^d \to \mathbb{R}^d$, the probability distribution for the random variable $\boldsymbol{x} = f(\boldsymbol{z})$ is given by

$$\mathcal{P}_{X,f}(\boldsymbol{x}) = \mathcal{P}_Z(\boldsymbol{z})|\det J_f(\boldsymbol{z})|^{-1} \,, \tag{10}$$

where $J_f = \partial f / \partial \boldsymbol{z}$ is the Jacobian of the differentiable transformation $f$. The full map $F$ produced by the NF architecture is composed from a sequence of $n$ such transformations $\boldsymbol{z} \equiv \boldsymbol{z}_0 \to \boldsymbol{z}_1 \equiv f_1(\boldsymbol{z}_0) \to \cdots \to \boldsymbol{x} \equiv \boldsymbol{z}_n \equiv f_n(\boldsymbol{z}_{n-1})$, as shown in fig. 7, with the final distribution given by

$$\mathcal{P}_X(\boldsymbol{x}) = \mathcal{P}_Z(\boldsymbol{z}) \prod_{i=1}^{n} |\det J_{f_i}(\boldsymbol{z}_{i-1})|^{-1} \,. \tag{11}$$

The NF architecture provides a continuous map from the latent space $Z$ to the target space $X$ and vice versa. In order to train the network to generate high fidelity mappings of samples from the latent distribution $\mathcal{P}_Z(\boldsymbol{z})$ to samples of the target distribution $\mathcal{P}_X(\boldsymbol{x})$ we

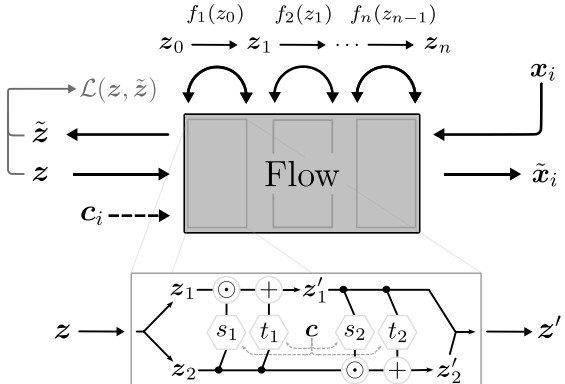

Figure 7: (top) Schematic of the input and output of the NF architecture. Here $\boldsymbol{x}_i$ and $\tilde{\boldsymbol{z}}$ represent, respectively, the input and output samples obtained from the network when mapping in the forward direction and $\boldsymbol{z}$ and $\tilde{\boldsymbol{x}}_i$ represent, respectively, input and output samples of the network obtained when traversing the network in the backward direction. The backward direction is used for event generation while the forward direction is used for training. The forward (backward) direction consists of a series of $n$ successive transformations $f_{i+1}(\boldsymbol{z}_i)$ ($f_i^{-1}(\boldsymbol{x}_i)$).

require a learning objective that will drive our model distribution $\mathcal{P}_X(\boldsymbol{x}; \boldsymbol{\theta})$ towards $\mathcal{P}_X(\boldsymbol{x})$. Given training samples $\boldsymbol{x}_a$ of $N$ data points, $\boldsymbol{x}_a = \{\boldsymbol{x}_1, \boldsymbol{x}_2, \ldots, \boldsymbol{x}_N\}$, with conditional labels $\boldsymbol{c}_a = \{\boldsymbol{c}_1, \boldsymbol{c}_2, \ldots, \boldsymbol{c}_N\}$, we use the minimization of the negative log likelihood as our learning objective,

$$
\begin{aligned}
\mathcal{L}_{\mathrm{NF}} &= \mathbb{E}_{\mathcal{P}_X(\boldsymbol{x}, \boldsymbol{c})} \left[ -\log \mathcal{P}_X(\boldsymbol{x}; \boldsymbol{\theta}, \boldsymbol{c}) \right] = -\sum_{a=1}^{N} \log \mathcal{P}_X(\boldsymbol{x}_a; \boldsymbol{\theta}, \boldsymbol{c}_a) \\
&= \sum_{a=1}^{N} \left\{ -\log \mathcal{P}_Z \left( F^{-1}(\boldsymbol{x}_a; \boldsymbol{\theta}, \boldsymbol{c}_a) \right) + \log |\det J_{F^{-1}}(\boldsymbol{x}_a; \boldsymbol{\theta}, \boldsymbol{c}_a)| \right\},
\end{aligned}
\tag{12}
$$

where $F(\boldsymbol{x}; \boldsymbol{\theta}, \boldsymbol{c})$ denotes the full network, parameterized by weights $\boldsymbol{\theta}$ and conditioned on labels $\boldsymbol{c}$. For a latent space sampled from a two-dimensional normal distribution, the loss function is given by

$$
\mathcal{L}_{\mathrm{NF}} = \sum_{a=1}^{N} \left\{ \frac{1}{2} ||F^{-1}(\boldsymbol{x}_a; \boldsymbol{\theta}, \boldsymbol{c}_a)||_2^2 - \log \left| \det J_F \left( F^{-1}(\boldsymbol{x}_a; \boldsymbol{\theta}, \boldsymbol{c}_a) \right) \right| \right\}.
\tag{13}
$$

where $|| \cdots ||_2^2$ denotes the squared $\ell^2$-norm. Because each operation is differentiable, the gradient of $\mathcal{L}_{\mathrm{NF}}$ with respect to each model parameter $\boldsymbol{\theta}$ may be computed using standard auto-differentiation software and optimized through stochastic gradient descent. Intuitively, the loss function in eq. (13) ensures that the latent variables obtained from mapping the training data samples through the network, *i.e.*, propagated from $\boldsymbol{x}_a \to \boldsymbol{z}_a$, are normally distributed.

For practical applications, the latent distribution $\mathcal{P}_Z(\boldsymbol{z})$ is chosen such that it can be easily evaluated and sampled, while the transformations $f_i$ are chosen such that (1) they are expressive enough to sufficiently approximate the transformation $\mathcal{P}_Z(\boldsymbol{z}) \to \mathcal{P}_X(\boldsymbol{x})$ and (2) they have computationally inexpensive Jacobians. For example, in the one-dimensional examples presented in section 4, we use a uniform latent distribution, $\boldsymbol{z} \sim \mathcal{U}_{[0,1]}$, and a mixture of Gaussian cumulative distribution functions as the transformations $f_i$. For the two-dimensional examples presented in section 3, we use a two-dimensional unit Gaussian

latent distribution, $z \sim \mathcal{N}(\vec{0}, \mathbb{I}_{2 \times 2})$, and real-valued non-volume preserving (real NVP) transformations, as implemented in the FREIA software library [28], for $f_i$. In the following subsections, we provide additional details regarding the architectures used in the one and two-dimensional models presented in the main text.

### B.1.1 One-dimensional normalizing flows

For one-dimensional distributions and a single map $f : \mathbb{R}^1 \to \mathbb{R}^1$, the transformation formula of eq. (10) can be rewritten as

$$\log \mathcal{P}_{X,f} = \log \mathcal{P}_Z(f(x)) + \log \left| \frac{\mathrm{d}f(x)}{\mathrm{d}x} \right| . \tag{14}$$

In one-dimension we can utilize a cumulative distribution function (CDF) as the invertible transformation $f$. Because CDFs are continuous, non-decreasing functions, they are guaranteed to have a unique inverse. Additionally, because CDFs satisfy

$$\lim_{x \to -\infty} \mathrm{CDF}(x) = 0 \text{ and } \lim_{x \to \infty} \mathrm{CDF}(x) = 1 , \tag{15}$$

a function that consists of a linear sum of CDFs is a CDF itself, *i.e.*,

$$\lim_{x \to \infty} \left( \sum_i w_i \mathrm{CDF}_i(x) \right) = \sum_i w_i \left( \lim_{x \to \infty} \mathrm{CDF}_i(x) \right) = \sum_i w_i = 1 , \tag{16}$$

as long as the weights $w$ are normalized such that the right most equality is true and

$$\lim_{x \to -\infty} \left( \sum_i w_i \mathrm{CDF}_i(x) \right) = \sum_i w_i \left( \lim_{x \to -\infty} \mathrm{CDF}_i(x) \right) = \sum_i w_i \times 0 = 0 . \tag{17}$$

In the main text we use a weighted linear mixture of $K$ Gaussian CDFs, $\Phi(x; \mu_i, \sigma_i)$, as the invertible transformation where the weights $w_i$, means $\mu_i$, and standard deviations $\sigma_i$ of each Gaussian component are tunable parameters learned by the network. This setup is commonly referred to as a Gaussian mixture model (GMM). The transformation and its derivative can be written explicitly as

$$f(x) = \sum_i^K w_i \Phi(x; \mu_i, \sigma_i) = \frac{1}{2} \sum_i^K w_i \left[ 1 + \mathrm{erf} \left( \frac{x - \mu_i}{\sqrt{2} \sigma_i} \right) \right] , \tag{18}$$

$$\frac{\mathrm{d}f(x)}{\mathrm{d}x} = \sum_i^K w_i \mathcal{N}(x; \mu_i, \sigma_i), \tag{19}$$

where we have used the fact that $\mathrm{d}\Phi(x; \mu_i, \sigma_i)/\mathrm{d}x = \mathcal{N}(x; \mu, \sigma)$.

Once a latent distribution $\mathcal{P}_Z$ is specified, the network can be trained by maximizing eq. (14)). After training, samples can be obtained in the inverse direction via inverse transform sampling. For $n$ successive transformations eq. (14), is modified to

$$\log \mathcal{P}_X(x) = \log \mathcal{P}_Z(F(x)) + \sum_{i=1}^n \log \left| \frac{\mathrm{d}f_i(x_{i-1})}{\mathrm{d}x} \right| . \tag{20}$$

To increase the flexibility of the network, we insert a total of $n-1$ intermediate non-linear functions $y_i$ between each transformation $f_i$ and $f_{i+1}$ such that the full transformation $F$ is given by $F = f_n(y_{n-1}(f_{n-1}(\cdots(y_1(f_1(x)))\cdots)))$ and the sum in eq. (20) runs from $i = 1, \ldots, 2n - 1$.

Specifically, we use a logit transformation defined as

$$y(x) = \text{logit}\left(\frac{\alpha}{2} + (1-\alpha)x\right), \text{ where } \text{logit}(x) = \log\frac{1}{1-x}, \tag{21}$$

with the derivative

$$\frac{\mathrm{d}y(x)}{\mathrm{d}x} = \frac{1-\alpha}{x(1-x)}. \tag{22}$$

Here, $\alpha$ is a hyperparameter of the network that is set to 0.01. The full architecture used in section 4 utilizes a uniformly distributed latent distribution $\mathcal{P}_Z \sim \mathcal{U}_{[0,1]}$ and $n = 5$ transformations $f_i$, where each transformation contains $K = 500$ Gaussian components. The learnable parameters are the Gaussian means and variances and the weights of the components, which are constrained to sum to unity per transformation.

### B.1.2 Two-dimensional flows

In section 3 we use two-dimensional real NVP transformations for $f_i$. Real NVP transformations consist of modular blocks containing two affine coupling layers. That is, given an input $\boldsymbol{z}_{i-1}$, the coupling block splits the input into two channels[8] $\boldsymbol{z}_{i-1} = \{\boldsymbol{z}_{i-1,1}, \boldsymbol{z}_{i-1,2}\}$ and applies a sequential affine transformation to each channel as follows

$$\begin{aligned}
\boldsymbol{z}_{i,1} &= \boldsymbol{z}_{i-1,1} \odot \exp\big(s_{i,1}\left(\boldsymbol{z}_{i-1,2}\right)\big) + t_{i,1}(\boldsymbol{z}_{i-1,2}), \\
\boldsymbol{z}_{i,2} &= \boldsymbol{z}_{i-1,2} \odot \exp\big(s_{i,2}\left(\boldsymbol{z}_{i,1}\right)\big) + t_{i,2}(\boldsymbol{z}_{i,1}),
\end{aligned} \tag{23}$$

where $s_{i,a}$ and $t_{i,a}$ are scale and translation transformation operators, respectively, parameterized by fully-connected multi-layer-perceptrons, while $\odot$ denotes the element-wise direct product.

Once passed through the coupling layer, the output $\boldsymbol{z}_i = \{\boldsymbol{z}_{i,1}, \boldsymbol{z}_{i,2}\}$ of the two channels is concatenated to the final output $f_i(\boldsymbol{z}_{i-1}) = \boldsymbol{z}_i$. The full architecture consists of $n$ sequential coupling blocks. Note that in the inverse direction,

$$\begin{aligned}
\boldsymbol{z}_{i,2} &= \big(\boldsymbol{z}_{i+1,2} - t_{i,2}\left(\boldsymbol{z}_{i+1,1}\right)\big) \odot \exp\big(-s_{i,2}\left(\boldsymbol{z}_{i+1,1}\right)\big), \\
\boldsymbol{z}_{i,1} &= \big(\boldsymbol{z}_{i+1,1} - t_{i,1}\left(\boldsymbol{z}_{i,2}\right)\big) \odot \exp\big(-s_{i,1}\left(\boldsymbol{z}_{i,2}\right)\big),
\end{aligned} \tag{24}$$

the $s_{i,a}$ and $t_{i,a}$ transformations are still evaluated in the forward direction and thus do not require a tractable inverse.

By construction, the Jacobian matrix $J_f$ for each coupling block is upper triangular, which allows for an efficient computation of its determinant

$$\det J_f(\boldsymbol{z}) = \det\frac{\partial f_{ij}}{\partial \boldsymbol{z}} = \det\begin{pmatrix} \text{diag}\left\{\exp\big(s_{i,1}(\boldsymbol{z}_{i,2})\big)\right\} & \cdots \\ 0 & \text{diag}\left\{\exp\big(s_{i,2}(\boldsymbol{z}_{i+1,1})\big)\right\} \end{pmatrix} \tag{25}$$

$$= \prod \exp\big(s_{i,1}(\boldsymbol{z}_{i,2})\big) \prod \exp\big(s_{i,2}(\boldsymbol{z}_{i+1,1})\big).$$

Because the transformations $s_{i,a}$ and $t_{i,a}$ can be arbitrarily complicated, the full architecture can be conditioned by concatenating labels $\boldsymbol{c}$ to the inputs of $s_{i,a}$ and $t_{i,a}$, i.e., $s_{i,a}(\boldsymbol{z}), t_{i,a}(\boldsymbol{z}) \to s_{i,a}(\boldsymbol{z},\boldsymbol{c}), t_{i,a}(\boldsymbol{z},\boldsymbol{c})$.

---

[8]Because we will deal with two-dimensional random variables, in our case both $\boldsymbol{z}_1$, $\boldsymbol{z}_2$ are one-dimensional. Which component of $\boldsymbol{z}$ is assigned as $\boldsymbol{z}_1$ or $\boldsymbol{z}_2$ is randomly chosen for each real NVP but is kept consistent over the complete dataset and stored for inference.

## B.2 Bayesian neural networks

When fitting a model to a data sample, it is often useful to understand the correlations and uncertainties related to the best-fit parameters. These uncertainties provide both information on the stability of the fit as well as information on the statistical variations within the data sample. Training a generative neural network is akin to a model fit, involving the optimization of network parameters to minimize a learning objective that produces samples matching the training dataset. As such, it is informative to understand the uncertainties associated with the network parameters. In deterministic neural networks, model parameters are single valued and remain fixed after training. There are a number of proposed methods for evaluating model uncertainties in deterministic networks, including the incorporation of drop-out layers [29], $k$-folding cross-validation [30], network ensemble averaging [31], *etc.*

However, these methods are either prescription-dependent, *e.g.*, a choice of the drop-out scheme, with no guarantee of comprehensive coverage, or require additional training cycles, making them computationally prohibitive for sufficiently complex networks. An alternative to these methods, which provides a systematic and statistically coherent assignment of uncertainties to model output, can be provided by Bayesian neural networks (BNNs) [32,33]. The major difference between deterministic neural networks and their Bayesian counterparts resides in the conversion of single-valued network parameters to parameters that are sampled according to a posterior, approximated as a product of normal distributions with mean and variance learned from the training dataset. In this context, the stability and uncertainty of model output is understood over an ensemble of samplings in network parameter space.

Consider a BNN whose goal is to accurately model the functional relationship $\boldsymbol{y} = f(\boldsymbol{x})$. The BNN is parameterized by model parameters $\boldsymbol{\theta}$ distributed before training according to a prior $\mathcal{P}(\boldsymbol{\theta})$ and the model output $f(\boldsymbol{x})$ is understood as a likelihood describing the probability $\mathcal{P}(\boldsymbol{y}|\boldsymbol{x}, \boldsymbol{\theta})$ of output $\boldsymbol{y}$, given the model parameters $\boldsymbol{\theta}$ and input $\boldsymbol{x}$. After training the model with a labeled dataset of $N$ pairs $\{(\boldsymbol{x}_n, \boldsymbol{y}_n)\}_{n=1}^N$, the probability of a particular output $\boldsymbol{y}$ for a given $\boldsymbol{x}$, the posterior predictive, may be written as a marginalization over the model parameter space

$$\mathcal{P}(\boldsymbol{y}|\boldsymbol{x}) = \int \mathrm{d}\boldsymbol{\theta}\, \mathcal{P}(\boldsymbol{y}|\boldsymbol{x}, \boldsymbol{\theta})\mathcal{P}(\boldsymbol{\theta}|\{(\boldsymbol{x}_n, \boldsymbol{y}_n)\}_{n=1}^N)\,, \tag{26}$$

where $\mathcal{P}(\boldsymbol{\theta}|\{(\boldsymbol{x}_n, \boldsymbol{y}_n)\}_{n=1}^N)$ represents the posterior distribution determined by the training data.

In practice, the actual form of $\mathcal{P}(\boldsymbol{\theta}|\{(\boldsymbol{x}_n, \boldsymbol{y}_n)\}_{n=1}^N)$ is analytically intractable and thus difficult to extract, although possible using Markov chain Monte Carlo techniques. Because of this, it is common practice to approximate this posterior through variational inference, where we approximate the posterior with a proposal distribution $\mathcal{Q}(\boldsymbol{\theta})$, typically chosen as a product of per parameter Gaussians, with all means and variances collectively denoted by $\boldsymbol{\phi}$. An accurate model of the posterior is one where the difference between $\mathcal{P}(\boldsymbol{\theta}|\{(\boldsymbol{x}_n, \boldsymbol{y}_n)\}_{n=1}^N)$ and $\mathcal{Q}(\boldsymbol{\theta}; \boldsymbol{\phi})$ is minimized. This can be achieved by minimizing the KL divergence

$$\min \mathrm{KL}\big(\mathcal{Q}(\boldsymbol{\theta}; \boldsymbol{\phi}), \mathcal{P}(\boldsymbol{\theta}|\{(\boldsymbol{x}_n, \boldsymbol{y}_n)\}_{n=1}^N)\big)\,, \tag{27}$$

where

$$\mathrm{KL}\big(\mathcal{Q}(\boldsymbol{\theta}; \boldsymbol{\phi}), \mathcal{P}(\boldsymbol{\theta}|\{(\boldsymbol{x}_n, \boldsymbol{y}_n)\}_{n=1}^N)\big) = -\int \mathrm{d}\boldsymbol{\theta}\, \mathcal{Q}(\boldsymbol{\theta}; \boldsymbol{\phi}) \log \frac{\mathcal{P}(\boldsymbol{\theta}|\{(\boldsymbol{x}_n, \boldsymbol{y}_n)\}_{n=1}^N)}{\mathcal{Q}(\boldsymbol{\theta}; \boldsymbol{\phi})}\,. \tag{28}$$

Bayes' theorem allows us to rewrite the intractable posterior

$$\mathcal{P}(\boldsymbol{\theta}|\{(\boldsymbol{x}_n, \boldsymbol{y}_n)\}_{n=1}^N) = \frac{\prod_{n=1}^N \mathcal{P}(\boldsymbol{y}_n|\boldsymbol{\theta}, \boldsymbol{x}_n)\mathcal{P}(\boldsymbol{\theta})}{\mathcal{P}(\{\boldsymbol{y}_{n'}\}_{n'=1}^N|\{\boldsymbol{x}_{n'}\}_{n'=1}^N)} \tag{29}$$

where $\mathcal{P}(\boldsymbol{y}_n|\boldsymbol{\theta}, \boldsymbol{x}_n)$ is the per-event likelihood and $\mathcal{P}(\boldsymbol{\theta})$ denotes the assigned prior on the model parameters. Introducing this into the KL divergence we obtain

$$\mathrm{KL}\big(\mathcal{Q}(\boldsymbol{\theta};\boldsymbol{\phi}), \mathcal{P}(\boldsymbol{\theta}|\{(\boldsymbol{x}_n, \boldsymbol{y}_n)\}_{n=1}^N)\big) =$$
$$= -\int \mathrm{d}\boldsymbol{\theta}\, \mathcal{Q}(\boldsymbol{\theta};\boldsymbol{\phi}) \log \frac{\prod_{n=1}^N \mathcal{P}(\boldsymbol{y}_n|\boldsymbol{\theta}, \boldsymbol{x}_n)\mathcal{P}(\boldsymbol{\theta})}{\mathcal{P}(\{\boldsymbol{y}_{n'}\}_{n'=1}^N|\{\boldsymbol{x}_{n'}\}_{n'=1}^N)\mathcal{Q}(\boldsymbol{\theta};\boldsymbol{\phi})} \tag{30}$$
$$= \log \mathcal{P}(\{\boldsymbol{y}_{n'}\}_{n'=1}^N|\{\boldsymbol{x}_{n'}\}_{n'=1}^N) - \mathcal{L}_{\mathrm{ELBO}}\,.$$

Because the KL divergence is non-negative and the evidence $\log \mathcal{P}$ is not a function of $\boldsymbol{\phi}$, the minimization in eq. (27) is achieved by maximizing the so-called evidence lower bound (ELBO) contribution, defined within the square brackets,

$$\mathcal{L}_{\mathrm{ELBO}} = \int \mathrm{d}\boldsymbol{\theta}\, \mathcal{Q}(\boldsymbol{\theta};\boldsymbol{\phi}) \left[\sum_{n=1}^N \log \mathcal{P}(\boldsymbol{y}_n|\boldsymbol{\theta}, \boldsymbol{x}_n) + \log \frac{\mathcal{P}(\boldsymbol{\theta})}{\mathcal{Q}(\boldsymbol{\theta};\boldsymbol{\phi})}\right]$$
$$= \sum_{n=1}^N \int \mathrm{d}\boldsymbol{\theta}\, \mathcal{Q}(\boldsymbol{\theta};\boldsymbol{\phi}) \log \mathcal{P}(\boldsymbol{y}_n|\boldsymbol{\theta}, \boldsymbol{x}_n) - \mathrm{KL}\big(\mathcal{Q}(\boldsymbol{\theta};\boldsymbol{\phi}), \mathcal{P}(\boldsymbol{\theta})\big) \tag{31}$$
$$\approx \frac{1}{M}\sum_{j=1}^M \sum_{n=1}^N \log \mathcal{P}(\boldsymbol{y}_n|\boldsymbol{\theta}_j, \boldsymbol{x}_n) - \mathrm{KL}\big(\mathcal{Q}(\boldsymbol{\theta};\boldsymbol{\phi}), \mathcal{P}(\boldsymbol{\theta})\big), \text{ where } \boldsymbol{\theta}_j \sim \mathcal{Q}(\boldsymbol{\theta};\boldsymbol{\phi})\,.$$

Above, we have approximated in the last line the expectation value of $\sum_{n=1}^N \log \mathcal{P}(\boldsymbol{y}_n|\boldsymbol{\theta}, \boldsymbol{x}_n)$ sampled over $\mathcal{Q}(\boldsymbol{\theta};\boldsymbol{\phi})$ as a summed average of $\sum_{n=1}^N \log \mathcal{P}(\boldsymbol{y}_n|\boldsymbol{\theta}, \boldsymbol{x}_n)$ evaluated at $M$ points of $\boldsymbol{\theta}$ distributed according to $\mathcal{Q}(\boldsymbol{\theta};\boldsymbol{\phi})$. Maximization of the ELBO loss $\mathcal{L}_{\mathrm{ELBO}}$ also minimizes the KL divergence in eq. (27), with the benefit that it requires no knowledge about the intractable distributions $\mathcal{P}(\boldsymbol{\theta}|\{(\boldsymbol{x}_n, \boldsymbol{y}_n)\}_{n=1}^N)$ and $\mathcal{P}(\{\boldsymbol{y}_{n'}\}_{n'=1}^N|\{\boldsymbol{x}_{n'}\}_{n'=1}^N)$. The first term in the ELBO loss drives the model to provide an accurate fit to the training data, while the second term acts as a regulator by weighting possible model parameters with a chosen prior $\mathcal{P}(\boldsymbol{\theta})$.

### B.3 Bayesian normalizing flows

Incorporating the Bayesian framework into the normalizing flow architecture, appendix B.1, only requires replacing the deterministic transformations $s_{i,a}$ and $t_{i,a}$ in eq. (23) with their Bayesian counterparts, *i.e.*, the BNNs. The full normalizing flow network architecture with BNN subnetworks is referred to as a Bayesian normalizing flow (BNF) [17]. The learning objective is equivalent to eq. (31) with the likelihood $\mathcal{P}(\boldsymbol{y}_n|\boldsymbol{\theta}_j, \boldsymbol{x}_n)$ now the NF model likelihood $\mathcal{P}_X(\boldsymbol{x};\boldsymbol{\theta}, \boldsymbol{c})$, *i.e.*, with the replacement $\boldsymbol{y} \to \boldsymbol{x}$ in the notation, and with no additional input measurements, only the model parameters $\boldsymbol{\theta}$ which determine $F(\boldsymbol{x}|\boldsymbol{\theta})$. The ELBO loss function in eq. (31) is therefore replaced by the following loss function,

which includes a minus sign to minimize rather than maximize,

$$
\begin{aligned}
\mathcal{L}_{\mathrm{BNF}} &= -\sum_{n=1}^{N} \mathbb{E}_{\boldsymbol{\theta} \sim \mathcal{Q}(\boldsymbol{\theta},\boldsymbol{\phi})} \left[ \log p_X^F(\boldsymbol{x}_n; \boldsymbol{\theta}, \boldsymbol{c}_n) \right] + \mathrm{KL}\big(\mathcal{Q}(\boldsymbol{\theta};\boldsymbol{\phi}), \mathcal{P}(\boldsymbol{\theta})\big) \\
&= -\sum_{n=1}^{N} \mathbb{E}_{\boldsymbol{\theta} \sim \mathcal{Q}(\boldsymbol{\theta},\boldsymbol{\phi})} \Bigg[ \log \mathcal{P}_Z\big(F^{-1}(\boldsymbol{x}_n; \boldsymbol{\theta}, \boldsymbol{c}_n)\big) \\
&\quad + \log\big|\det J_F\left(F^{-1}(\boldsymbol{x}_n; \boldsymbol{\theta}, \boldsymbol{c}_n)\right)\big|\Bigg] + \mathrm{KL}\big(\mathcal{Q}(\boldsymbol{\theta};\boldsymbol{\phi}), \mathcal{P}(\boldsymbol{\theta})\big) \\
&\approx -\sum_{n=1}^{N} \frac{1}{M} \sum_{m=1}^{M} \Bigg\{ \log \mathcal{P}_Z\big(F^{-1}(\boldsymbol{x}_n; \boldsymbol{\theta}_m, \boldsymbol{c}_n)\big) \\
&\quad + \log\big|\det J_F\left(F^{-1}(\boldsymbol{x}_n; \boldsymbol{\theta}_m, \boldsymbol{c}_n)\right)\big|\Bigg\} + \mathrm{KL}\big(\mathcal{Q}(\boldsymbol{\theta};\boldsymbol{\phi}), \mathcal{P}(\boldsymbol{\theta})\big).
\end{aligned}
\tag{32}
$$

We assume a two dimensional standard normal latent space, along with both a Gaussian prior $\mathcal{P}(\boldsymbol{\theta})$ and a Gaussian variational distribution $\mathcal{Q}(\boldsymbol{\theta})$, *i.e.*,

$$
\mathcal{P}(\boldsymbol{\theta}; \boldsymbol{\mu}_{\mathcal{P}}, \boldsymbol{\sigma}_{\mathcal{P}}) = \mathcal{N}(\boldsymbol{\theta}; \boldsymbol{\mu}_{\mathcal{P}}, \boldsymbol{\sigma}_{\mathcal{P}}), \qquad \mathcal{Q}(\boldsymbol{\theta}; \boldsymbol{\mu}_{\mathcal{Q}}, \boldsymbol{\sigma}_{\mathcal{Q}}) = \mathcal{N}(\boldsymbol{\theta}; \boldsymbol{\mu}_{\mathcal{Q}}, \boldsymbol{\sigma}_{\mathcal{Q}}).
\tag{33}
$$

The KL divergence for $D$ parameters is then given by

$$
\mathrm{KL}\big(\mathcal{Q}(\boldsymbol{\theta};\boldsymbol{\phi}), \mathcal{P}(\boldsymbol{\theta})\big) = \sum_{d=1}^{D} \frac{\sigma_{\mathcal{Q},d}^2 - \sigma_{\mathcal{P},d}^2 + (\mu_{\mathcal{Q},d} - \mu_{\mathcal{P},d})^2}{2\sigma_{\mathcal{P},d}^2} + \log \frac{\sigma_{\mathcal{P},d}}{\sigma_{\mathcal{Q},d}}.
\tag{34}
$$

Choosing a standard normal prior, $\mu_{\mathcal{P},d} = 0$, $\sigma_{\mathcal{P},d} = 1$ for $d = 1, \ldots, D$, leaves us with our final BNF loss function

$$
\begin{aligned}
\mathcal{L}_{\mathrm{BNF}} &= \sum_{n=1}^{N} \frac{1}{M} \sum_{m=1}^{M} \left\{ \frac{||F^{-1}(\boldsymbol{x}_n; \boldsymbol{\theta}_m, \boldsymbol{c}_n)||_2^2}{2} - \log\big|\det J_F\left(F^{-1}(\boldsymbol{x}_n; \boldsymbol{\theta}_m, \boldsymbol{c}_n)\right)\big| \right\} \\
&\quad + \sum_{d=1}^{D} \left[ \frac{1}{2}(\sigma_{\mathcal{Q},d}^2 + \mu_{\mathcal{Q},d}^2 - 1) - \log \sigma_{\mathcal{Q},d} \right].
\end{aligned}
\tag{35}
$$

Above, the expression in the first line represents the same NF loss function as $\mathcal{L}_{\mathrm{NF}}$ in eq. (13), but now averaged over $M$ samplings of the network parameters. In practice, it is typical to set $M = 1$ in order to reduce the computational cost of training, with the understanding that the constraints imposed on the Jacobian structure will ensure that the mapping to and from the latent space will remain stable with non-divergent gradients. The second term represents the optimization of the individual network weight distributions to best replicate the intractable posterior $\mathcal{P}(\boldsymbol{\theta}|\{\boldsymbol{x}_n\}_{n=1}^N)$ via a product of Gaussians with tunable $\boldsymbol{\mu}_{\mathcal{Q}}$ and $\boldsymbol{\sigma}_{\mathcal{Q}}$ that are not too far away from the prior values 0 and 1.

## B.4    Interpreting BNF ensembles

After training the BNF network one obtains a probabilistic model defined over a distribution of network parameters $\boldsymbol{\theta}$. This distribution should, in principle, contain information about both the model stability and uncertainties due to the training dataset. That is, the BNF defines an envelope of possible NFs defined by different values of $\boldsymbol{\theta}$ distributed according to probability distribution $\mathcal{Q}(\boldsymbol{\theta})$. Concrete values of $\boldsymbol{\theta}$ give a particular realization of the NF, *i.e.*, a particular map between the latent and target spaces. All observable

quantities should be computed as averages over many samplings of the network parameters $\boldsymbol{\theta}$.

To assess how the BNF encodes the training uncertainties into the learned densities, we follow ref. [34][9] and consider the statistical distribution of an observable $g$ defined over a generated dataset, $g(\{\boldsymbol{x}_i\}_{i=1}^{N_{\text{gen}}})$, where $N_{\text{gen}}$ denotes the sample size of the generated data. For a given set of parameters $\boldsymbol{\theta}$, $g(\{\boldsymbol{x}_i\}_{i=1}^{N_{\text{gen}}})$ will have a likelihood $\mathcal{P}(g(\{\boldsymbol{x}_i\}_{i=1}^{N_{\text{gen}}})|\boldsymbol{\theta})$. An example of such an observable $g$ is, *e.g.*, the number of times an emission falls into a given bin, in which case the likelihood $\mathcal{P}(g(\{\boldsymbol{x}_i\}_{i=1}^{N_{\text{gen}}})|\boldsymbol{\theta})$ is a Poisson distribution with the average rate

$$\langle N_{\text{bin}}\rangle_{\boldsymbol{\theta}} = N_{\text{gen}} \int_{\boldsymbol{x}\in\text{bin}} \mathrm{d}\boldsymbol{x}\, \mathcal{P}(\boldsymbol{x}|\boldsymbol{\theta})\,, \tag{36}$$

and the usual related variance

$$\sigma^2_{\text{bin}|\boldsymbol{\theta}} = \langle N_{\text{bin}}\rangle_{\boldsymbol{\theta}}\,. \tag{37}$$

The integral in $\boldsymbol{x}$ can be approximated by sampling from $\mathcal{P}(\boldsymbol{x}|\boldsymbol{\theta})$. The likelihood $\mathcal{P}(g(\{\boldsymbol{x}_i\}_{i=1}^{N_{\text{gen}}})|\boldsymbol{\theta})$ will in turn determine the mean value of the observable, $\mathbb{E}_{X[N_{\text{gen}}]|\boldsymbol{\theta}}[g]$, and its variance, $\sigma^2_{X[N_{\text{gen}}]|\boldsymbol{\theta}}[g]$, where $X[N_{\text{gen}}]$ is the space of all possible datasets of size $N_{\text{gen}}$. In general, these will not be analytic functions of $\boldsymbol{\theta}$ and will need to be determined numerically.

Since one needs to take into account all possible networks this affects the expectation value and the variance of $g$,

$$
\begin{aligned}
\mathbb{E}_{X[N_{\text{gen}}],\Theta}[g] &= \int \mathrm{d}\boldsymbol{\theta}\, \mathcal{P}(\boldsymbol{\theta}|\{x_j\}_{j=1}^{N_{\text{train}}})\mathbb{E}_{X[N_{\text{gen}}]|\boldsymbol{\theta}}[g]\,, \\
\sigma^2_{X[N_{\text{gen}}],\Theta} &= \int \mathrm{d}\boldsymbol{\theta}\, \mathcal{P}(\boldsymbol{\theta}|\{x_j\}_{j=1}^{N_{\text{train}}})\mathbb{E}_{X[N_{\text{gen}}]|\boldsymbol{\theta}}[(g - \mathbb{E}_{X[N_{\text{gen}}],\Theta}[g])^2] \\
&= \int \mathrm{d}\boldsymbol{\theta}\, \mathcal{P}(\boldsymbol{\theta}|\{x_j\}_{j=1}^{N_{\text{train}}})\big(\mathbb{E}_{X[N_{\text{gen}}]|\boldsymbol{\theta}}[g^2] - \mathbb{E}_{X[N_{\text{gen}}]|\boldsymbol{\theta}}[g]^2 \\
&\quad + \big(\mathbb{E}_{X[N_{\text{gen}}]|\boldsymbol{\theta}}[g] - \mathbb{E}_{X[N_{\text{gen}}],\Theta}[g]\big)^2\big) \\
&= \sigma^2_{\text{gen}} + \sigma^2_{\text{train}}\,,
\end{aligned}
\tag{38}
$$

where $\mathbb{E}_{X[N_{\text{gen}}],\Theta}$ is the expectation value over the whole distribution of $\boldsymbol{\theta}$, $N_{\text{train}}$ denotes the sample size of the training data, and $\sigma^2_{\text{gen}}$ and $\sigma^2_{\text{train}}$ are given in the third and the fourth lines of eq. (38), respectively. For the relevant example where $g$ is the number of times an emission falls into a given bin or region, we have

$$
\begin{aligned}
\mathbb{E}_{X[N_{\text{gen}}],\Theta}[N_{\text{bin}}] &= \int \mathrm{d}\boldsymbol{\theta}\, \mathcal{P}(\boldsymbol{\theta}|\{x_j\}_{j=1}^{N_{\text{train}}})\langle N_{\text{bin}}\rangle_{\boldsymbol{\theta}} = \langle N_{\text{bin}}\rangle\,, \\
\sigma^2_{\text{gen}} &= \int \mathrm{d}\boldsymbol{\theta}\, \mathcal{P}(\boldsymbol{\theta}|\{x_j\}_{j=1}^{N_{\text{train}}})\sigma^2_{\text{bin}|\boldsymbol{\theta}} = \langle N_{\text{bin}}\rangle\,, \\
\sigma^2_{\text{train}} &= \int \mathrm{d}\boldsymbol{\theta}\, \mathcal{P}(\boldsymbol{\theta}|\{x_j\}_{j=1}^{N_{\text{train}}})\left(\langle N_{\text{bin}}\rangle_{\boldsymbol{\theta}} - \langle N_{\text{bin}}\rangle\right)^2\,.
\end{aligned}
\tag{39}
$$

We observe how $\sigma_{\text{gen}}$ is the uncertainty of a Poisson distributed random variable whose expected rate is given by $\langle N_{\text{bin}}\rangle$. That is, in this case $\sigma_{\text{gen}}$ will not depend on the characteristics of the posterior except for its mean, and thus does not include any training uncertainties. This is reflected in the fact that although the posterior depends on the

---

[9]See Ref. [35] for another application of BNNs to HEP where the uncertainties are explicitly decomposed in terms of aleatoric and epistemic uncertainties with the help of a bicephalous regression network. Although the methodology is different, the decomposition of aleatoric and epistemic presented in Ref. [35] is equivalent to the decomposition presented here in terms of generation and training uncertainties.

training dataset of size $N_{\text{train}}$, $\sigma_{\text{gen}}$ itself depends exclusively on the size of the generated dataset of size $N_{\text{gen}}$. In general, the statistical variance of a randomly distributed $g$ whose distribution is determined by the BNF is captured by $\sigma_{\text{gen}}$ and does not vanish, even if we collapse the posterior $\mathcal{P}(\boldsymbol{\theta}|\{x_j\}_{j=1}^{N_{\text{train}}})$ to a delta function. For instance, for the example where $g$ is the number of times and emission falls into a given bin, we have

$$
\begin{aligned}
\sigma_{\text{gen}}^2 &= \int \mathrm{d}\boldsymbol{\theta}\, \delta\big(\boldsymbol{\theta} - \boldsymbol{\theta}^{\text{MAP}}(\{x_j\}_{j=1}^{N_{\text{train}}})\big) \big(\mathbb{E}_{X[N_{\text{gen}}]|\boldsymbol{\theta}}[g^2] - \mathbb{E}_{X[N_{\text{gen}}]|\boldsymbol{\theta}}[g]^2\big) \\
&= \int \mathrm{d}\boldsymbol{\theta}\, \delta\big(\boldsymbol{\theta} - \boldsymbol{\theta}^{\text{MAP}}(\{x_j\}_{j=1}^{N_{\text{train}}})\big)\langle N_{\text{bin}}\rangle_{\boldsymbol{\theta}} \\
&= \langle N_{\text{bin}}\rangle_{\boldsymbol{\theta}^{\text{MAP}}} = N \int_{\boldsymbol{x}\in\text{bin}} \mathrm{d}\boldsymbol{x}\, \mathcal{P}(\boldsymbol{x}|\boldsymbol{\theta}^{\text{MAP}}) \,,
\end{aligned}
\tag{40}
$$

where $\boldsymbol{\theta}^{\text{MAP}}$ is the maximum a posteriori (MAP) estimate. That is, we recover the Poisson uncertainty $\sigma_{\text{gen}}^2 = \langle N_{\text{bin}}\rangle$. Again, we note that although $\boldsymbol{\theta}^{\text{MAP}}$ depends on the training dataset of size $N_{\text{train}}$, the Poisson uncertainty of the observable itself reflects the size of the studied dataset of size $N_{\text{gen}}$.

The fourth line of eq. (38), $\sigma_{\text{train}}^2$, captures how the expected values of the observables change due to uncertainties in $\boldsymbol{\theta}$ and does vanish if we collapse the posterior to a delta function. The larger $\sigma_{\text{train}}^2$ is, the larger the variations in possible networks sampled from the posterior and the larger the set of $\boldsymbol{\theta}$ parameter values resulting in outputs consistent with the training dataset.

## B.5    Architecture impact on the BNF uncertainty

In section 3, we showed how the BNF captures the training uncertainty. However, this uncertainty also depends on the BNF architecture. In fig. 8 we show the uncertainty variations as a function of the number of nodes per hidden layer in the BNF, while fig. 9 shows two examples of learned $p_z$ distributions and the values of the associated BNF parameters $\boldsymbol{\mu}_{\mathcal{Q}}$ and $\boldsymbol{\sigma}_{\mathcal{Q}}$ (see appendix B.4 for the details about the notation). We especially highlight the model with 32 nodes, which was used in the main text for both fig. 4 and fig. 5. In fig. 8 this model is denoted with a star. Although the results in fig. 8 and fig. 9 do not represent an exhaustive scan, since the number of hidden layers per module and number of modules remained fixed to 2 modules with 4 layers each, we can nevertheless begin to characterize the behavior we obtain for alternative models.

When the number of nodes is too low, the BNF model is simultaneously both inflexible (the $p_z$ distribution does not match the training data) and overly certain. The learned standard deviations $\boldsymbol{\sigma}_{\mathcal{Q}}$ for the posterior are too small, collapsing the posterior for $\boldsymbol{\mu}_{\mathcal{Q}}$ to a delta function. The BNF then always samples very similar NFs, whose parameters are given by the learned means $\boldsymbol{\mu}_{\mathcal{Q}}$. This behavior can be seen in the two upper panels in fig. 9, which shows the results for a learned BNF model with only 1 node per hidden layer.

Conversely, when the number of nodes is large enough, and we appropriately regularize, the learned average distribution matches much better the data, and the posterior is broad enough to appropriately capture the model uncertainties. We observe how $\boldsymbol{\sigma}_{\mathcal{Q}}$ is now large and thus allows a higher variation on the sampled $\boldsymbol{\theta}$. The learned means $\boldsymbol{\mu}_{\mathcal{Q}}$ should be non-zero to yield a non-trivial MAP distribution, although the specific values are hard to analyze due to the inherent complexity of the neural network. Additionally, the model shows a subset of weights which are effectively removed by having very low mean and variance. This indicates redundant parameters which could be removed by pruning or by a refinement of the architecture. The lower two panels in fig. 9 demonstrate this behavior for a learned model with 32 nodes per hidden layer.

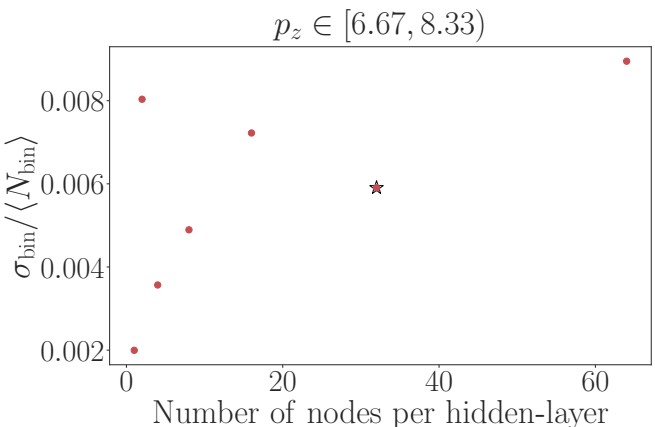

Figure 8: Relative total uncertainty for the most populous bin, as a function of the model architecture. We scan over the number of nodes per hidden layer and obtain the total uncertainty for the same bin as a function of said nodes per hidden layer. The chosen bin is the one with the largest expected event count for the model with 32 nodes per hidden layer, the default architectural choice, denoted with a star.

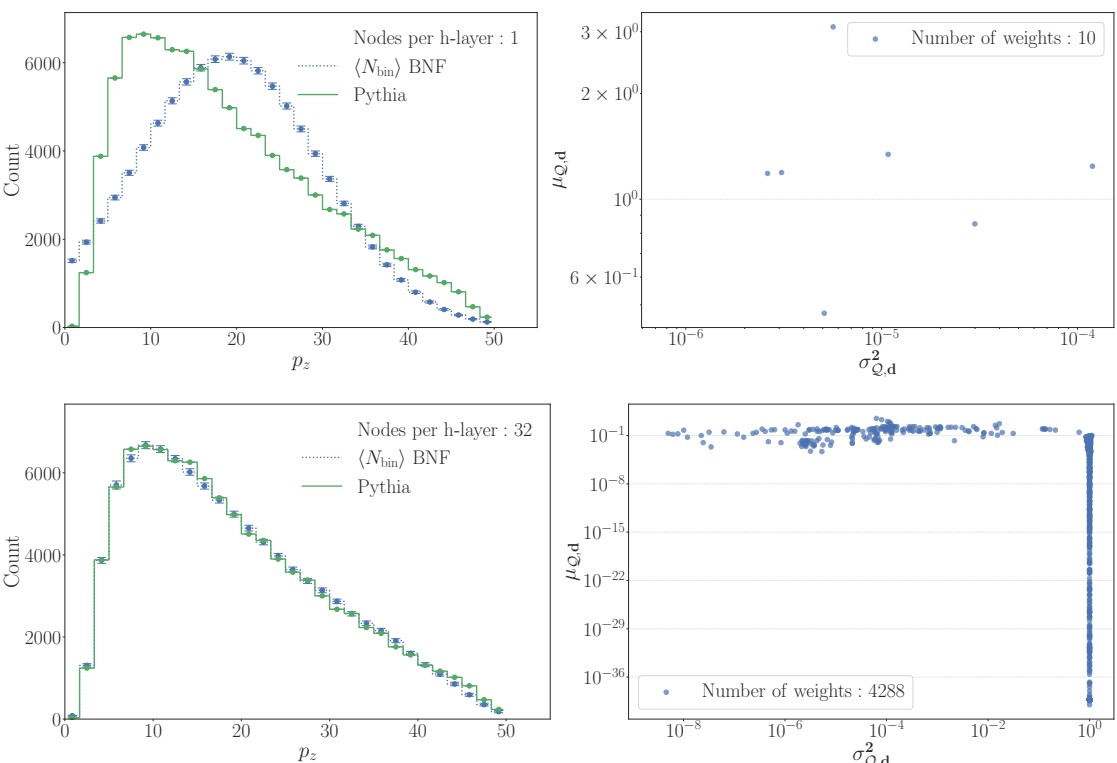

Figure 9: Impact of the model architecture on the quality of the model. The top (bottom) row corresponds to a model with 1 (32) nodes per hidden layer. (left) Comparison of the training dataset and the BNF predictions. (right) Visualization of the inferred parameters of the BNF posterior distribution for the model weights, the means $\boldsymbol{\mu}_{\mathcal{Q}}$ and standard deviations $\boldsymbol{\sigma}_{\mathcal{Q}}$ of the approximate Gaussian posterior.

We observe that the uncertainty as displayed in fig. 8 is a good reflection of training quality but not necessarily a good metric for model selection. This is exemplified by comparing the 16 node model to the 32 node model. We observe that the former has a larger uncertainty despite providing a reasonable description of data. In this case, the uncertainty increases because the model provides a poor prediction in the example bin. Since the model is both sufficiently expressive and well trained, it recognizes this mis-modeling and reflects it in the increased uncertainty. If we were interested in lighter models, we could choose the 16 node model at the expense of a slightly larger uncertainty for the largest bin. In general, model selection should take into account all bins, as well as other considerations, such as the model size. In this work, however, we were not interested in selecting the optimal model but in selecting a descriptive model with sensible uncertainties.

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
