# Peer review of "Towards a data-driven model of hadronization using normalizing flows"

_SciPost Physics_

## Round 1 · Referee Report · Anonymous (Referee 1) · 2024-2-20

Strengths

1- Presents a novel way to sample systematic variation by utilizing the probability distribution given by normalizing flows.

2- Enables tuning of the normalizing flow models to the experimental distributions.

3- Bayesian normalizing flow allows for estimating modeling uncertainties.

4- Detailed explanation of the methodology and demonstrates with dedicated examples

5- Provides interpretation of results and model behavior from case studies.

Weaknesses

1- Examples provided are oversimplified. They are only one-dimensional space. It is unclear if the proposed method will be able to learn the correlation between different variables in the multi-dimension scenario.

2- The proposed method is claimed to be more computation efficient but lacks benchmark comparison to the other existing methods, in terms of either computing time or model accuracy.

3- In the examples, it is shown that the estimated uncertainties using the bayesian method can cover the non-closure between model prediction and the target distribution, but there is no rigorous way to quantify how accurate these estimated uncertainties are.

4- In the case the model is under-fitting, there doesn't seem to be a way in the paper to estimate the uncertainty caused by the under-fitting.

Report

The paper generally meets the journal's acceptance criteria, with several minor changes requested, and improvements that could be made.

Requested changes

1- In my understanding, the paper in ref [10] has the same goal of determining microscopic dynamics from macroscopic observables, and it uses the observable (hadron kinematics as opposed to the hadron multiplicity that is considered in this paper) that is available at the experimental level. I suggest splitting the mentioning of ref [9] and ref [10], and highlighting the differences between this paper and ref [10] and potential advantages the proposed method in this paper can bring.

2- The last paragraph in section 3 should be moved to section 4.

3- Add a few sentences to discuss the multi-dimension scenario and the ability of the model to learn the correlation between variables.

4- Provide benchmarking of the timing and compare with other existing methods.

5- Discuss about the accuracy of the estimated uncertainties using the bayesian method. And discuss about whether the discrepancy between prediction and experimental distribution is caused by the modeling uncertainty or by the under-fitting of the model.

6- The y-axis in the left panel of fig. 3 reads PDF. Are these the direct likelihood given by the normalizing flows? Or they are the distribution sampled from the normalizing flows? There should not be a direct PDF of the target distribution, so the target distribution should be the distribution sampled from the event generator. In either case, please clarify.

7- It is not clear how the N_bin bins are divided in the right panel of fig. 4. Please clarify it.

  • validity: high
  • significance: good
  • originality: top
  • clarity: high
  • formatting: excellent
  • grammar: excellent

Author:  Tony Menzo  on 2024-04-23  [id 4440]

(in reply to Report 1 on 2024-02-20)

We thank the referee for their detailed reading of our manuscript and constructive comments. We have made amendments and additions within the manuscript to address the requested changes.

Weaknesses

  1. We intentionally used the 1D example to introduce the MAGIC method, in order to simplify the notation and the discussion. We have also performed limited tests with multi-dimensional microscopic models and only encountered the expected issue of not being able to solve the inverse problem when there was not enough macroscopic information given. We agree with the referee that further work on the MAGIC method is warranted, and we plan to perform more dedicated tests when this will be used in more complex problems in future works. To make the reader aware of potential issues we have expanded the relevant discussion on p. 9.
  2. We thank the referee for this comment. We have added the rough order of magnitude speed-up estimates for the case of our toy simulation on p.8.
  3. We agree with the reviewer that a rigorous way to assess the systematic uncertainties related to the (in)correctness of the hadronization models is lacking. We did, however, explore the issues of under- and over-fitting in appendix B.5 and we have added a more detailed discussion in the main text. We believe that implementing a more rigorous set of closure tests to assess model correctness lies beyond the scope of this work, given that the model was already shown to converge properly. However, this is an important topic and deserves careful study in the future.
  4. We thank the referee for pointing out the incomplete discussion. We have expanded our discussion in the main text on how such an uncertainty can be assigned by performing a model comparison between different architectures much like what was done in appendix B.5.

Requested changes

  1. The referee is correct in stating that ref. [10] has similar goals of extracting microscopic information about the hadronization models using only measurable quantities. We have added two clarifying sentences in the introduction. It would also be interesting to perform a more detailed comparison between the two approaches (once they are developed enough for the comparison to be meaningful). We have mentioned this possibility on p. 9 in the main text.
  2. We thank the reviewer for the suggestion and have rearranged the text accordingly.
  3. We have added additional details on the multi-dimensional scenario in the second to last paragraph of section 4. We have tested and expect MAGIC training to generalize to correlated multi-dimensional distributions assuming the macroscopic distributions used during training are sensitive to the correlations exhibited by the base NF. In general, we expect multi-dimensional fine-tunings with MAGIC training to require more oversight during training to ensure proper coverage during model exploration and reweighting.
  4. We have included a footnote in the second to last paragraph of the introduction and changed the text in the sixth paragraph of section 4 to clarify that the comparison and gain in computation time is meant to refer to the savings between re-simulating events versus re-weighting events, the former taking on the order of minutes per thousands of events versus seconds per thousands of events. While the NF architecture is able to generate events more quickly than the model introduced in our previous work [8], this is not the main focus of the present study and both models are still significantly slower than modern event generators. We leave attempts at producing generative-ML-based models of hadronization with sampling speeds comparable to modern event generators for future work.
  5. We have added to the main text a brief discussion regarding under- and over-fitting of the model, and model correctness in general. We reference explicit techniques for model comparison in a Bayesian framework [29, 30] and the architecture scan performed in appendix B.5. In particular, we also discuss how the posterior uncertainties reflect a lack of significant under-fitting. Of course, the model is not a perfect approximation of the experimental distribution but it is not so biased towards a wrong model in such a way that the uncertainties are artificially low. A more thorough closure test can quantify the degree of compatibility between model and data. However, we believe such an analysis is not warranted in this case given the performance of the model.
  6. Figure 3 shows the learned PDF as defined in eq. (20) of appendix B.1.1 obtained by feeding in an ordered list of points over the domain of the distribution. By construction, the 1D-NF is automatically normalized to unity.
  7. Figure 4 is a scatter plot illustrating the relationship between the counts of bins from the left plot and the relative error in the corresponding bin. It compares the (blue) BNF estimate of the relative uncertainty, $\sigma_{\text{bin}} / \langle N_{\text{bin}} \rangle$, and the (red) relative generated uncertainty, $\sigma_{\text{gen}} / \langle N_{\text{bin}} \rangle$, for each bin. We have clarified this in the figure caption.

Anonymous on 2024-05-31  [id 4533]

(in reply to Tony Menzo on 2024-04-23 [id 4440])
Category:
correction
suggestion for further work

With regard to reply (4) of the requested changes we would like to amend the reply to the following -

We have changed the text in the sixth paragraph of section 4 to clarify that the comparison and gain in computation time is meant to refer to the savings between re-simulating events versus re-weighting events, the former taking on the order of minutes per thousands of events versus seconds per thousands of events. We have also included a footnote in the second to last paragraph of the introduction to clarify that while the NF architecture is able to generate events more quickly than the model introduced in our previous work [8] (the event generation time improves by roughly 1.8 times), both models are still significantly slower than modern event generators. We emphasize that this is not the main focus of the present study. We leave attempts at producing generative-ML-based models of hadronization with sampling speeds comparable to modern event generators for future work.

---

## Round 1 · Referee Report · Anonymous (Referee 2) · 2024-2-29

Strengths

  • Well written, clear and compact style
  • Methods described clearly
  • Meaningful examples
  • Applicability and limitations well discussed
  • Technical details described in useful appendices

Weaknesses

  • application yet limited, but paper marks an important step towards a meaningful application of machine learning in hadronization.

Report

The authors report on a method to describe the hadronization of a quark-antiquark pair with a machine learning based model utilizing normalizing flows. The study is based on a simplified variant of the Lund string fragmentation model that only produces pions as hadrons. As such the authors show that the introduced model can learn the hadronization of the Lund string fragmentation and reproduce the detailed microscopic description of this and even allows to show improvements of the detailed underlying fragmentation function. Furthermore, the method allows to study the uncertainties and sensitivities in terms of the Lund fragmentation parameters. As the authors have motivated this study, this allows for important insight into the hadronization model itself and may provide hints to possible improvement of the modeling and understanding of the microscopic mechanisms.

The paper is very well written and all details and ideas are laid out transparently and can be followed by the reader. The authors provide meaningful figures that describe the performance of the introduced methodology. At the same time the limitations are clearly marked and possible avenues for future research are shown. This paper marks a step towards more detailed studies of hadronization models that may indeed lead to a better understanding of non-perturbative physics in the hadronization stage of high energy particle collisions.

The paper is recommended for publication in SciPost without changes.
  • validity: top
  • significance: top
  • originality: high
  • clarity: top
  • formatting: perfect
  • grammar: perfect

Author:  Tony Menzo  on 2024-04-23  [id 4441]

(in reply to Report 2 on 2024-02-29)

We thank the referee for their detailed reading of the manuscript and constructive comments.

---

## Editorial Decision

resubmitted